# *Stem-OB:* Generalizable Visual Imitation Learning with Stem-Like Convergent Observation through Diffusion Inversion

**Kaizhe Hu**[123*]  **Zihang Rui**[1*]  **Yao He**[4]  **Yuyao Liu**[1]  **Pu Hua**[123]  **Huazhe Xu**[123]

[1] Tsinghua University  [2] Shanghai Qi Zhi Institute  [3] Shanghai AI Lab  [4] Stanford University

`hukaizhe22@mails.tsinghua.edu.cn, huazhe_xu@mail.tsinghua.edu.cn`

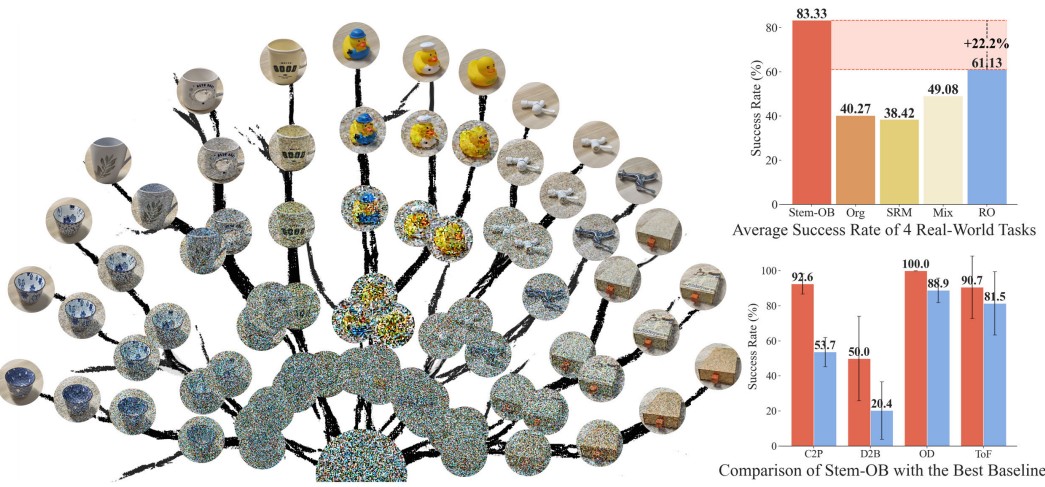

Figure 1: **Left:** The tree of *Stem-OB* inversion is composed of different objects progressively inverted through a diffusion inversion process. Moving downward alone the tree's branches, objects of different textures, appearances, and categories gradually get closer, eventually converging into the same root of Gaussian noise, where they are completely indistinguishable. **Right:** Real-world tasks success rate, where *Stem-OB* showcases a significant improvement.

## ABSTRACT

Visual imitation learning methods demonstrate strong performance, yet they lack generalization when faced with visual input perturbations like variations in lighting and textures. This limitation hampers their practical application in real-world settings. To address this, we propose **Stem-OB** that leverages the inversion process of pretrained image diffusion models to suppress low-level visual differences while maintaining high-level scene structures. This image inversion process is akin to transforming the observation into a shared representation, from which other observations also stem. *Stem-OB* offers a simple yet effective plug-and-play solution that stands in contrast to data augmentation approaches. It demonstrates robustness to various unspecified appearance changes without the need for additional training. We provide theoretical insights and empirical results that validate the efficacy of our approach in simulated and real settings. *Stem-OB* shows an exceptionally significant improvement in real-world robotic tasks, where challenging light and appearance changes are present, with an average increase of **22.2%** in success rates compared to the best baseline. See our website for more videos.

## 1 INTRODUCTION

Visual Imitation Learning (IL), where an agent learns to mimic the behavior of the demonstrator by learning a direct mapping from visual observations to low-level actions, has gained popularity in recent real-world robot tasks (Chi et al., 2023; Zhao et al., 2023; Wang et al., 2023a; Chi et al.,

---

*Indicates equal contribution.

2024; Ze et al., 2024). Despite the versatility demonstrated by visual IL, learned policies are often brittle and fail to generalize to unseen environments, even minor perturbations such as altering the lighting conditions or the texture of the object may lead to failure of the learned policy (Xie et al., 2023; Yuan et al., 2024b). The underlying reason is that the high-dimensional visual observation space is redundant with virtually infinite variations in appearance that are irrelevant to the task and hard to generalize.

As human beings, we can easily manipulate objects that have different appearances. For example, we can pick up a coffee cup regardless of its color, texture, or the lighting condition of the room. This is partially because our visual system is capable of abstracting the high-level semantics of the scene, such as the silhouette of the object, the structure and arrangement of different objects, etc in a hierarchical manner (Hochstein & Ahissar, 2002), effectively merging scenes with perceptual differences to similar "meta" observations.

Augmentation techniques such as Spectrum Random Masking (SRM) (Huang et al., 2022) and Mixup (Zhang et al., 2018) remove details from observations to encourage the model to focus on structural features; however, they lack the ability to distinguish between low-level and high-level features. It is preferable if we can sweep the photometrical differences while maintaining the high-level structure for the scene. Achieving this requires a semantic understanding of the observations, and naively perturbing the data with Gaussian noise can lead to irreversible information loss.

Pretrained large image diffusion models, such as Stable Diffusion (Rombach et al., 2022; Esser et al., 2024), embed essential world knowledge for visual understanding. Apart from synthesizing new images from random noise, these models are capable to perform a reverse procedure called inversion (Song et al., 2022), which converts an image back to the space of random noises. A recent study (Yue et al., 2024) indicates that this inversion process selectively eliminates information from the image. Rather than uniformly removing information from different semantic hierarchies, it will push those images with similar structures closer in the early stages of the inversion process. Inversion is like the reprogramming of a differentiated cell back to a stem cell, which bears the totipotency to differentiate into any cell type. This characteristic aligns perfectly with our will of enhancing the robustness and generalizability of visual IL algorithms to visual variations. To distill such property into a visual IL policy, we propose an imitation learning pipeline which applies diffusion inversion to the visual observations. We name our method *Stem-OB* to highlight the similarity between the inversed observation and the stem cell in biology, as illustrated in Figure 1.

To be specific, our method is as simple as inverting the image for reasonable steps before sending them to the downstream visual IL algorithms. The number of steps is chosen empirically to balance removing irrelevant details without erasing essential high-level information. From this perspective, our approach fundamentally distinguishes from generative augmentation methods, which aim to enrich the training dataset with more unseen objects and appearances (Yu et al., 2023; Mandlekar et al., 2023). Moreover, *Stem-OB* is indifferent to many unspecified appearance changes, in contrast to augmentation-based methods that must concentrate on a few selected types of generalization, thereby introducing inevitable inductive biases.

We provide theoretical analysis and a user study to support our claim that *Stem-OB* can effectively merge scenes with perceptual differences to similar "stem observations". Empirical study demonstrates the effectiveness of our approach in a variety of simulated and real-world tasks and a range of different perturbations. *Stem-OB* proves to be particularly effective in real-world tasks where appearance and lighting changes hamper the other baselines, establishing an overall improvement in the success rate of **22.2%**. What's better, no inference time inversion is required for *Stem-OB* to take effect, making the deployment of our method virtually free of computational cost.

## 2 RELATED WORKS

### 2.1 VISUAL IMITATION LEARNING AND GENERALIZATION

Visual Imitation Learning (VIL) is a branch of Imitation learning (IL) that focuses on learning action mappings from visual observations. It typically follows two approaches: directly imitating expert policies, as in behavior cloning and DAgger (Ross et al., 2011), or inferring a reward function that aligns the agent's behavior with expert demonstrations, like inverse reinforcement learning (Ng &

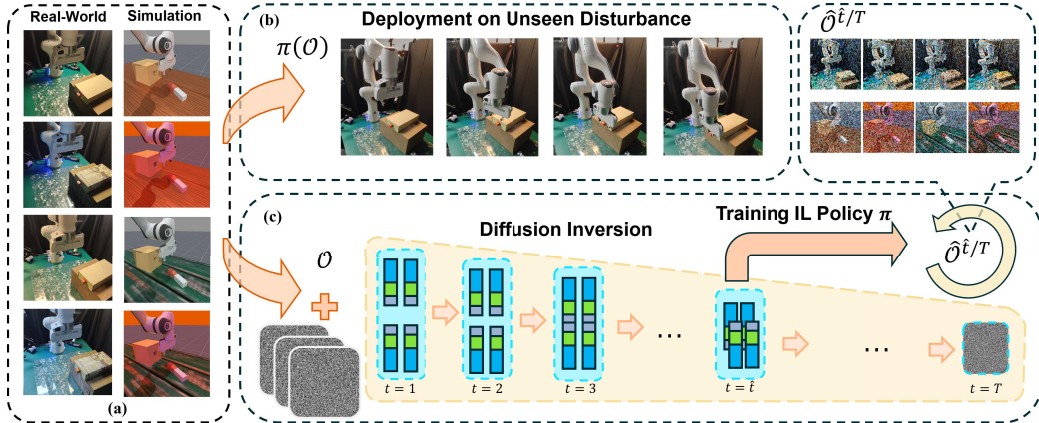

Figure 2: **Overview of *Stem-OB*: (a)**. *Stem-OB* has been evaluated in both real-world and simulated environments. **(b)**. The trained visual IL policies are directly applied to the original observation space $\mathcal{O}$, demonstrating robustness to unseen environmental disturbances. **(c)**. We train the visual IL policy $\boldsymbol{\pi}$ on the diffusion-inversed latent space $\hat{\mathcal{O}}^{\hat{t}/T}$, where $\hat{t}$ denotes a specific inversion step out of a total of $T$. Each composite rectangle in the diffusion inversion process, made up of three smaller sections, represents the latent vector of an image, with the smaller section depict the finer attributes (gray). During the inversion process, finer attributes converge earlier than coarser ones.

Russell, 2000) and GAIL (Ho & Ermon, 2016). The former approach, favored in recent works due to its scalability and practicality in complex, real-world tasks, has led to several advancements. Notable methods include Diffusion Policy (Chi et al., 2023), which leverages a diffusion model (Ho et al., 2020) to maximize the likelihood of expert actions, and Action Chunk Transformer (Zhao et al., 2023) that uses a Transformer (Vaswani, 2017).

To enhance the generalization and robustness of visual imitation learning algorithms, various approaches have been explored (Cetin & Celiktutan, 2021). For instance, Li et al. (2023) leverage trajectory and step level similarity through an estimated reward function, while Wan et al. (2023) improve visual robustness by separating task-relevant and irrelevant dynamics models before applying the GAIL framework (Ho & Ermon, 2016). Zhang et al. (2023a) use mutual information constraints to create compact representations that generalize to unseen environments. However, these approaches are not directly applicable to methods like diffusion policy, which focus on imitation without reward functions or dynamics models. Zheng et al. (2023) propose to filter extraneous action subsequence, yet their focus is not on visual perturbations. Most relevant to our setting, several works in robust visual reinforcement learning have explored adding noise in image or frequency domain to improve generalizability (Huang et al., 2022; Lee & Hwang, 2024; Yuan et al., 2024a), however, they lack the semantic understanding of the augmentation process.

## 2.2 INVERSION OF DIFFUSION MODELS AND ITS APPLICATION

Diffusion model inversion aims to recover an initial noise distribution from a given image, enabling image reconstruction from the same noise via backward denoising. A common approach is DDIM inversion (Song et al., 2022), which estimates the previous noise using predictions from the current diffusion step, though the approximation will introduce cumulative errors. To address this issue, several methods employ learnable text embeddings (Mokady et al., 2023; Miyake et al., 2023), fixed-point iterations (Pan et al., 2023; Garibi et al., 2024), or gradient-based techniques (Samuel et al., 2024) to refine the result. Another approach, based on the stochastic DDPM scheduler (Ho et al., 2020), reconstructs noisy images at each diffusion step (Huberman-Spiegelglas et al., 2024; Brack et al., 2024). In contrast to the DDIM inversion methods, the noise $\tilde{\epsilon}_t$ of each step is statistically independent, making them ideal for our application since we can obtain the noisy image of a certain step without the need to recover other steps, greatly reducing the calculation cost.

Diffusion Inversion is the crucial part of diffusion-based image editing methods (Meng et al., 2021; Kawar et al., 2023), which typically involves first inverting the diffusion process to recover the noise latent, then denoise the latent with desired editing conditions. Recent works also explore to apply attention control over the denoising process to improve the fidelity of the edited image Hertz et al.

(2022); Tumanyan et al. (2023), and have shown promising application in robot learning tasks Gao et al. (2024); Ju et al. (2025); Zhu et al. (2024). Beyond that, inversion is also used in tasks like concept extraction Huang et al. (2023) and personalization Gal et al. (2022). Most recently, Wang & Chen (2024) proposed the use of diffusion inversion to interpolate between image categories to improve classification performance. And Wang et al. (2023b) uses diffusion inversion to erase out sub-optimal trajectories from the dataset.

## 3  PROBLEM DEFINITION

Given a dataset of observation $\mathcal{O}$ and action $\mathcal{A}$ pairs, the goal of VIL is to learn a policy $\pi_\theta(\mathcal{A}|\mathcal{O})$ that maps observations to actions. The policy is typically parameterized by a neural network with parameters $\theta$, and is trained to minimize the negative log-likelihood of the actions.

To achieve the goal of generalizing to unseen environments, we seek a method to preprocess or transform the observations $\mathcal{O}$ such that task-irrelevant details are suppressed while preserving the high-level semantic structure that is critical for the task. The problem can be transformed into learning a transformation $\mathcal{T}$ as the input to the policy $\pi_\theta(\mathcal{A}|\mathcal{T}(\mathcal{O}))$, where $\mathcal{T}(\mathcal{O})$ is the transformed observation emphasizing high-level semantics while removing irrelevant details.

## 4  PRELIMINARY

We begin by outlining the fundamentals of Diffusion Inversion. A diffusion model operates with two passes: a backward denoising pass, which generates an image from noise, and a forward pass, where noise is incrementally added to an image until it becomes pure Gaussian noise. This forward process is a Markov chain that starts with $\boldsymbol{x}_0$, and gradually adds noise to obtain latent variables $\boldsymbol{x}_1, \boldsymbol{x}_2, ..., \boldsymbol{x}_T$. Each step in this process is a Gaussian transition following the common form

$$\boldsymbol{x}_t = \sqrt{\alpha_t}\boldsymbol{x}_{t-1} + \sigma_t\boldsymbol{\epsilon}_t \sim \mathcal{N}(\boldsymbol{x}_t|\sqrt{\alpha_t}\boldsymbol{x}_t, \sigma_t^2\boldsymbol{I}) \tag{1}$$

where $\alpha_t \in (0, 1)$ represents the scheduler parameter at each step $t$, while $\sigma_t$ characterizes the variance of the Gaussian noise $\boldsymbol{\epsilon}_t$ introduced at each step. In Denoising Diffusion Probabilistic Models (DDPM) (Ho et al., 2020), $\sigma_t = \sqrt{1 - \alpha_t}$. Consequently, equation Eq. (1) can be reformulated as Eq. (2) by applying the cumulative product $\bar{\alpha}_t = \prod_{i=1}^{t}\alpha_i$

$$\boldsymbol{x}_t = \sqrt{\bar{\alpha}_t}\boldsymbol{x}_0 + \sqrt{1 - \bar{\alpha}_t}\boldsymbol{\epsilon}_t \sim \mathcal{N}(\boldsymbol{x}_t|\sqrt{\bar{\alpha}_t}\boldsymbol{x}_{t-1}, (1 - \bar{\alpha}_t)\boldsymbol{I}) \tag{2}$$

Diffusion inversion is similar to the forward process in that they both maps an image to a noise, however, inversion tries to preserve the image's information and obtain the specific noise that can reconstruct the image during a backward denoising process.

**DDPM inversion.** We follow the DDPM inversion proposed in Huberman-Spiegelglas et al. (2024),

$$\boldsymbol{x}_t = \sqrt{\bar{\alpha}_t}\boldsymbol{x}_0 + \sqrt{1 - \bar{\alpha}_t}\tilde{\boldsymbol{\epsilon}}_t \tag{3}$$

The DDPM inversion we consider here differs slightly from Eq. (2), as $\tilde{\boldsymbol{\epsilon}}_t \sim \mathcal{N}(\boldsymbol{0}, \boldsymbol{I})$ are mutually independent distributions, in contrast to the highly correlated $\boldsymbol{\epsilon}_t$ in Eq. (2). As mentioned by Huberman-Spiegelglas et al. (2024), the independence of $\tilde{\boldsymbol{\epsilon}}_t$ results in a sequence of latent vectors where the structures of $\boldsymbol{x}_0$ are more strongly imprinted into the noise maps. An error reduction step is conducted in reverse order after the diffusion forward process to improve image reconstruction accuracy during the denoising process:

$$\boldsymbol{z}_t = \boldsymbol{x}_{t-1} - \hat{\mu}(\boldsymbol{x}_t)/\sigma_t, \quad \boldsymbol{x}_{t-1} = \hat{\mu}(\boldsymbol{x}_t) + \sigma_t\boldsymbol{z}_t \tag{4}$$

**DDIM inversion.** We follow the DDIM inversion proposed in Song et al. (2022), where at each forward diffusion step

$$\boldsymbol{x}_t = \sqrt{\frac{\bar{\alpha}_t}{\bar{\alpha}_{t-1}}}\boldsymbol{x}_{t-1} + \left(\sqrt{\frac{1}{\bar{\alpha}_t} - 1} - \sqrt{\frac{1}{\bar{\alpha}_{t-1}} - 1}\right)\boldsymbol{\epsilon}_{\boldsymbol{\theta}}(\boldsymbol{x}_{t-1}, t, \mathcal{C}) \tag{5}$$

Note that the noise $\epsilon_{\boldsymbol{\theta}}(\boldsymbol{x}_{t-1}, t, \mathcal{C})$ is now generated by a network trained to predict the noise based on the previous vector $\boldsymbol{x}_{t-1}$ and the text embedding $\mathcal{C}$ which, in our case, is $\emptyset$.

## 5 METHOD

In this section, we introduce the intuition and implementation of our framework. We first propose the intuition of applying inversion on observations through theoretical analysis based on attribute loss, a diffusion-based measurement of image semantic similarity. Then, we conduct an illustrative experiment and a user study to validate our intuition. Finally, we explain how to practically implement *Stem-OB* and incorporate diffusion inversion into a visual imitation learning framework.

### 5.1 INTUITION DEVIATION BY ATTRIBUTE LOSS

Intuitively, as the diffusion inversion process moves forward, a source image and another variation of it become increasingly indistinguishable. The variation here could be low-level changes like lightning conditions, but also includes semantic changes such as replacing an object. If there are two different variations, we want to show that as the inversion step increases, the pair with minor alterations will become indistinguishable sooner than the pair with larger and structural changes. We borrow the definition of attribute loss from Yue et al. (2024) to quantify the semantic overlapping of the two images at time step $t$ during a inversion process:

$$loss(\boldsymbol{x}_0, \boldsymbol{y}_0, t) = \frac{1}{2}\text{OVL}(q(\boldsymbol{x}_t|\boldsymbol{x}_0), q(\boldsymbol{y}_t|\boldsymbol{y}_0)) \tag{6}$$

where $\boldsymbol{x}_0$ and $\boldsymbol{y}_0$ are the latent variables of the two images, and OVL is the overlapping coefficient quantifying the overlapping area of two probability density functions. For an inversion process where each step follows a Gaussian transition, it takes the form $\boldsymbol{x}_t = \sqrt{\bar{\alpha}_t}\boldsymbol{x}_0 + \sigma\epsilon \sim \mathcal{N}(\sqrt{\bar{\alpha}_t}\boldsymbol{x}_0, \sigma^2\mathbf{I})$. The OVL can be further calculated as the overlapping area of two Gaussian distributions, i.e.,

$$loss(\boldsymbol{x}_0, \boldsymbol{y}_0, t) = \frac{1}{2}\Big[1 - \text{erf}(\frac{||\sqrt{\bar{\alpha}_t}(\boldsymbol{y}_0 - \boldsymbol{x}_0)||}{2\sqrt{2}\sigma})\Big] \tag{7}$$

where erf is the error function, which is strictly increasing. Given a source image $\boldsymbol{x}_0$ and its variations $\hat{\boldsymbol{x}}_0$ and $\tilde{\boldsymbol{x}}_0$, with $\tilde{\boldsymbol{x}}_0$ undergoing a larger variation than $\hat{\boldsymbol{x}}_0$, the following conclusion can be easily observed under the same diffusion scheduling:

$$\tau(\boldsymbol{x_0}, \hat{\boldsymbol{x}}_0, \rho) < \tau(\boldsymbol{x_0}, \tilde{\boldsymbol{x}}_0, \rho), \ s.t. ||\hat{\boldsymbol{x}}_0 - \boldsymbol{x}_0|| < ||\tilde{\boldsymbol{x}}_0 - \boldsymbol{x}_0|| \tag{8}$$

Here, $\tau(\boldsymbol{x}_0, \boldsymbol{y}_0, \rho) = \inf\{t > 0 \mid loss(\boldsymbol{x}_0, \boldsymbol{y}_0, t) > \rho\}$ represents the earliest step where the loss between $\boldsymbol{x}_0$ and $\boldsymbol{y}_0$ exceeds the threshold $\rho$, and $||\cdot||$ measures the difference between an image and its variation. Eq. (8) provides a theoretical grounding for our intuition: images with fine-grained attribute changes tend to become indistinguishable sooner than those with coarse-grained modifications under identical diffusion schedules.

We can further derive the attribute loss for DDPM inversion

$$loss_{DDPM}(\boldsymbol{x}_0, \boldsymbol{y}_0, t) = \frac{1}{2}\Big[1 - \text{erf}(\frac{||\sqrt{\bar{\alpha}_t}(\boldsymbol{y}_0 - \boldsymbol{x}_0)||}{2\sqrt{2(1 - \bar{\alpha}_t)}})\Big] \tag{9}$$

Additionally, we derive that the attribute loss for DDIM inversion exhibits a similar form under certain assumptions. The detailed derivation can be found in Appendix A.2.

$$loss_{DDIM}(\boldsymbol{x}_0, \boldsymbol{y}_0, t) = \frac{1}{2}\left[1 - \text{erf}\left(\frac{||(\boldsymbol{y}_0 - \boldsymbol{x}_0)||}{2\sqrt{2\sum_{i=1}^{t}\frac{1}{\bar{\alpha}_i}\left(\sqrt{\frac{1}{\bar{\alpha}_i} - 1} - \sqrt{\frac{1}{\bar{\alpha}_{i-1}} - 1}\right)^2}}\right)\right] \tag{10}$$

Because $\bar{\alpha}_t \in (0, 1)$ is strictly decreasing, the attribute loss tends to increase as the time step increases. Furthermore, as discussed in Yue et al. (2024), this attribute loss is equivalent to how likely the DM falsely reconstruct $\boldsymbol{x}_t$ sampled from $q(\boldsymbol{x}_t|\boldsymbol{x}_0)$ closer to $\boldsymbol{y}_0$ instead of $\boldsymbol{x}_0$, and vise versa.

## 5.2 ILLUSTRATIVE EXPERIMENT

In Sec. 5.1, we made a key assumption that semantically similar images are closer in latent space of the diffusion models, leading to the conclusion that such images exhibit higher attribute loss at a given inversion step (Eq. (8)). While this assumption is backed by recent study on the zero-shot semantic correspondence ability on diffusion latents (Tang et al., 2023; Zhang et al., 2023b), we further conducted an illustrative experiment to further support it. To validate this assumption, we used a set of images, denoted as $\mathcal{I}$, from the real-world task variations described in Sec. 6.1.1. Specifically, we selected the 4 generalization objects in the real-world tasks from 5 categories and calculated the pairwise distance between images on their latent representations, both for intra-category and cross-category image pairs. The results are presented in Tab. 1, where each entry represents the average distance between two categories. It is clear that the diagonal entries indicating intra-category similarity exhibit lower loss compared to cross-category images, which justifies our claim.

We then conducted a user study to validate that similar images exhibit higher attribute loss at a given inversion step. We recruited 51 participants and presented each with image pairs from $\mathcal{I}$ after specific steps of diffusion inversion. The inversion steps were systematically sampled from 15 to 45, out of a total 50 steps inversion and with intervals of 5. Each participant was asked to determine whether the image pairs depicted the same object, and we recorded the proportion of incorrect responses.

Table 1: By-category image semantic similarity.

| Categories | Bowl | Cup | Drawer | Duck | Faucet |
|---|---|---|---|---|---|
| Bowl | 156.65 | 172.20 | 172.81 | 172.82 | 167.73 |
| Cup | 172.20 | 154.63 | 165.43 | 167.83 | 167.37 |
| Drawer | 172.81 | 165.43 | 144.63 | 161.82 | 161.57 |
| Duck | 172.82 | 167.83 | 161.82 | 140.51 | 147.00 |
| Faucet | 167.73 | 167.37 | 161.57 | 147.00 | 145.34 |

The experimental results in Fig. 3 show that at inversion step $t_1 = 20$, the incorrect response rate within the same category starts to increase. In contrast, the error rate for objects from different categories only started to rise at the inversion step $t_2 = 35$. This indicates that inversion makes objects within the same category harder to distinguish earlier than affecting the distinction between objects from different categories. Therefore, these results support our claim that similar images exhibit higher attribute loss at a given inversion step.

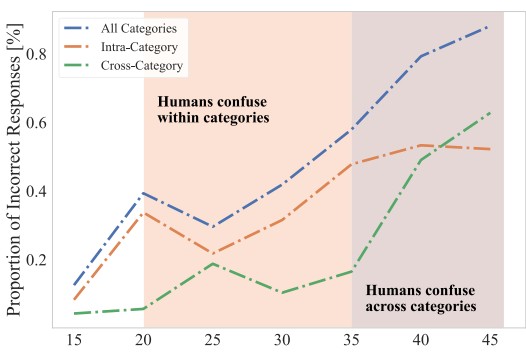

Figure 3: User study confusion proportions.

## 5.3 IMPLEMENTATION OF STEM-OB

Following the theoretical analysis and experimental validation, we propose a practical implementation of our framework. Our method applies diffusion inversion to converge different observations during training, and we find that the model gains robustness improvement in the original observation space during test time. The detailed implementation is described below.

**Training.** The training process begins by applying diffusion inversion to each observation. We define the following partial diffusion inversion process for an observation $\boldsymbol{o}_i \in \mathcal{O}$:

$$\hat{\boldsymbol{o}}_i^{t/T} = \boldsymbol{f}(\boldsymbol{o}_i, t, T) \tag{11}$$

Here, $\hat{\boldsymbol{o}}_i^{t/T} \in \hat{\mathcal{O}}^{t/T}$ denotes the observation after $t$ out of $T$ steps of diffusion inversion from the original observation $\boldsymbol{o}_i \in \mathcal{O}$. The function $\boldsymbol{f}(\cdot)$ applies $t$ steps of diffusion inversion to $\boldsymbol{o}$ using any inversion methods. We select the DDPM inversion method for its efficiency and effectiveness in our experiments, and the selection of $t$ and total inversion step $T$ is discussed in Sec. 6.5. The visual imitation learning algorithms are then trained on the inversion-altered space $\hat{\mathcal{O}}^{t/T}$. Note that we empirically find the error reduction step of DDPM inversion in Eq. (4) is not significant for the performance, so we omit it and approximate the partial inversion process with regard to Eq. (3) only. In this way, we avoid the full inversion process or reverse-order corrections that involve extensive Diffusion Model inference. This approach significantly reduces preprocessing time with minimal performance impact, achieving an average time for preprocessing of 0.2s per image.

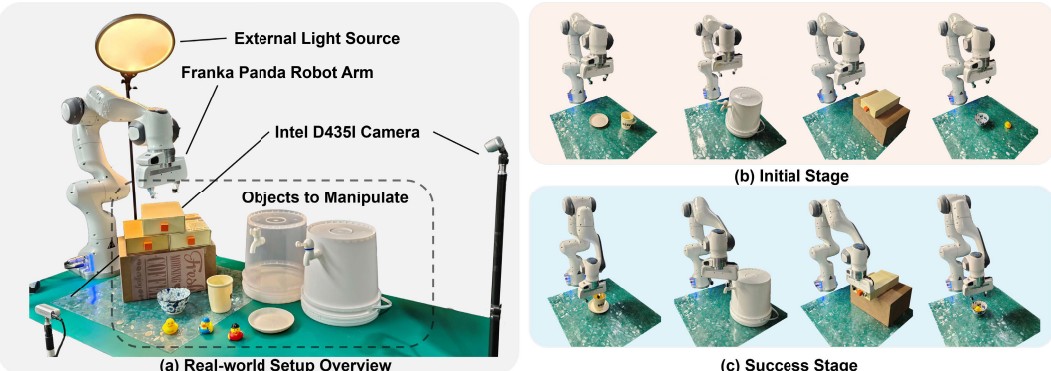

Figure 4: **Realworld Setup**: (a) Overview of the whole setup. (b)(c) These tasks are performed by the robot in a real-world environment, from left to right: *Cup2Plate*, *Turn on Faucet*, *Open Drawer*, and *Duck2Bowl*. The figure showcases the initial and final states of the tasks.

**Testing.** Despite training is entirely conducted on the inversion-altered space, the imitation learning algorithm of our choice showcases surprising zero-shot adaptation ability to the original observation space. We adhere to the original observation space $\mathcal{O}$ during testing, which means essentially no changes are made to the downstream visual imitation learning algorithms. This approach demonstrates improved robustness to environmental variations without any inference-time overhead, and is suitable for any other test-time augmentation techniques.

## 6 EXPERIMENTS

In this section, we present a comprehensive set of experiments using Diffusion Policy (DP) (Chi et al., 2023) as our visual imitation learning backbone. Since Stable Diffusion is pretrained on real-world images (Chi et al., 2023; Rombach et al., 2022), and a increasing interest in deploying visual imitation learning in real-world scenarios in the community (Paolo et al., 2024), we focus primarily on evaluating *Stem-OB* on real-world tasks. We extend the testing to simulated environments as well for further benchmarking. To assess the robustness of our method against visual appearance variations, we design experiments featuring different object textures and lighting conditions. We compare *Stem-OB* against several baselines to validate its effectiveness.

### 6.1 EXPERIMENT SETUP

#### 6.1.1 REAL-WORLD TASKS

We conduct real-world experiments using a Franka Emika Panda Arm, and two RealSense D435I cameras positioned at different angles for RGB inputs. The setup is shown in Fig. 4. Our experiments focus on four tasks, as described below:

*Cup to Plate* (C2P): The robot arm picks up a cup and places it on a plate, with variation introduced by changing tablecloth patterns.
*Duck to Bowl* (D2B): The robot grasps a toy duck and places it into a bowl, with variations introduced by altering the duck's appearance.
*Open Drawer* (OD): The robot arm grabs a drawer handle and pulls it open, with variations introduced by modifying the drawer's visual characteristics.
*Turn on Faucet* (ToF): The robot arm turns on a faucet, with variations introduced by altering the appearance of the faucet and bucket.

In addition to the visual variations mentioned above, all four tasks above involve changes in *lighting conditions*, i.e., cool vs. warm light. All the variations only happen during testing time, with the training set contain only a basic setting. The object locations in training set are randomly initialized within a specified area, and 100 demonstrations are collected per task. For testing, nine predefined target positions are used. Further details of the environmental variations can be found in Appendix C.1.

### 6.1.2 SIMULATION TASKS

Our simulation experiments consider different tasks within two frameworks: a photorealistic simulation platform SAPIEN 3 (Xiang et al., 2020) and a less realistic framework MimicGen (Mandlekar et al., 2023).

**SAPIEN 3** provides physical simulation for robots, rigid bodies, and articulated objects. It delivers high-fidelity visual and physical simulations that closely approximate real-world conditions. We leverage the ManiSkill 3 dataset (Gu et al., 2023; Tao et al., 2024), collected on SAPIEN 3, for benchmarking. Specifically, we select four tasks from ManiSkill for evaluation:

*PushCube*: The robot arm pushes a cube to a target location.
*PegInsertionSide*: The robot arm inserts a peg into a hole.
*PickCube*: The robot arm picks up a cube and places it on a target location.
*StackCube*: The robot arm stacks one cube over the other.

During testing, we vary the background and lighting conditions in all tasks to generate different visual appearances. 50 episodes are tested for each setting. Details of the environmental variations can be found in Appendix C.2.

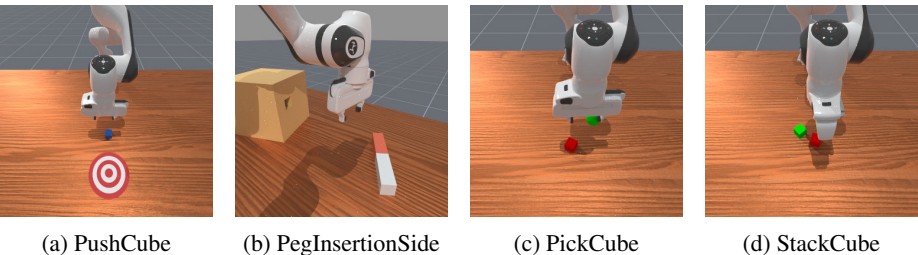

(a) PushCube          (b) PegInsertionSide          (c) PickCube          (d) StackCube

Figure 5: **SAPIEN 3 environments.** The figure showcases the visual appearance and task configuration of each setting.

**MimicGen** is a system for generating large diverse datasets from a small number of human demonstrations. We utilize the benchmark to evaluate the performance of our approach and conduct experiments on a variety of tasks: *MugCleanup*, *Threading*, *ThreePieceAssembly* and *Coffee*. MimicGen offers numerous task variations, each characterized by different initial distributions. We adopt the default initial distribution (D0) for both training data and test environments. For evaluation, we employ a single image as the input to the policy, using 500 samples out of a total of 1000 demos for training. We alter the table texture and object appearances to create different test environments. 300 episodes are tested for each setting of all the tasks. Detailed descriptions of the environmental variations can be found in Appendix C.3.

## 6.2 BASELINES

We compare *Stem-OB* with several data augmentation methods aimed at improving the generalizability of visual imitation learning. The baselines include **SRM** (Huang et al., 2022), which modifies images by adding random noise in the frequency domain, **Mix-Spectrum** (Mix) (Lee & Hwang), which enhances the agent's focus on semantic content by blending original images with randomly selected reference images from the same dataset in the frequency domain, and **original images** (Org) without any modification. Additionally, as highlighted in (Yuan et al., 2024b), **Random Overlay** (RO) improves generalization in real-world experiments by blending original images with random real-world photos in the image domain. Therefore, we include an additional study using Random Overlay in our real-world experiments.

## 6.3 EXPERIMENT RESULTS

### 6.3.1 REAL-WORLD EXPERIMENTS

In this section, we report the average success rate across all predefined test positions for each task. As shown in Tab. 2, *Stem-OB* consistently outperforms the baseline models in all settings, demon-

Table 2: **Evaluation of real-world experiments.** Train.: evaluations in the same settings as the training dataset. Gen.: evaluations under different visual perturbations for generalizability analysis. All: evaluations including both Train. and Gen. The tasks are *C2P*, *D2B*, *OD*, and *ToF*. We report the mean of the success rate (%) over corresponding settings for each task, and the best results are highlighted in **bold**.

| Task \ | *C2P* | | | *D2B* | | | *OD* | | | *ToF* | | |
|---|---|---|---|---|---|---|---|---|---|---|---|---|
| Algorithm | Train. | Gen. | All | Train. | Gen. | All | Train. | Gen. | All | Train. | Gen. | All |
| *Stem-OB* | **89.0** | **93.3** | **92.6** | **78.0** | **44.4** | **50.0** | **100.0** | **100.0** | **100.0** | **100.0** | **88.9** | **90.7** |
| Org | **89.0** | 77.8 | 79.6 | 56.0 | 13.3 | 20.4 | 89.0 | 13.3 | 25.9 | **100.0** | 22.2 | 35.2 |
| SRM | 56.0 | 73.3 | 70.4 | 44.0 | 8.9 | 14.8 | 89.0 | 15.6 | 27.8 | 89.0 | 31.1 | 40.7 |
| Mix | **89.0** | 55.6 | 61.1 | 44.0 | 22.2 | 25.9 | 33.0 | 35.6 | 35.2 | **100.0** | 68.9 | 74.1 |
| RO | 56.0 | 53.3 | 53.7 | 44.0 | 15.6 | 20.4 | 89.0 | 88.9 | 88.9 | 78.0 | 82.2 | 81.5 |

Table 3: **Evaluations in simulated environments.** We compare the performance of *Stem-OB*, Org, SRM, and Mix in simulated environments. The left side of the table shows the results on the SAPIEN 3 environment, while the right side is about the MimicGen environment. The results are reported as the mean and standard deviation (only for MimicGen) of the success rate(%) over their own test settings. The specific success rates are listed in Appendix E. The performance of *Stem-OB* is highlighted in orange for better visibility. The best performance of each task is highlighted in **bold**.

| Task \ | SAPIEN 3 | | | | MimicGen | | | |
|---|---|---|---|---|---|---|---|---|
| Algorithm | *PushCube* | *PegInsertionSide* | *PickCube* | *StackCube* | *MugCleanup* | *Threading* | *ThreePieceAssembly* | *Coffee* |
| *Stem-OB* | **99.1** | 0.4 | **25.1** | **27.2** | $19.5_{\pm 4.2}$ | $16.4_{\pm 3.1}$ | $13.1_{\pm 3.1}$ | **$50.5_{\pm 4.1}$** |
| Org | 61.8 | 0.0 | 8.6 | 4.3 | $16.4_{\pm 3.4}$ | $18.1_{\pm 2.5}$ | **$14.8_{\pm 2.3}$** | $43.3_{\pm 4.5}$ |
| SRM | 62.5 | 0.0 | 8.4 | 0.7 | $15.4_{\pm 3.0}$ | **$25.8_{\pm 4.6}$** | $13.6_{\pm 2.5}$ | $39.4_{\pm 2.9}$ |
| Mix | 97.2 | **0.6** | 12.9 | 8.2 | **$22.3_{\pm 3.6}$** | $18.9_{\pm 2.1}$ | $14.2_{\pm 2.2}$ | $41.8_{\pm 4.0}$ |

strating superior generalization capability. Under the training conditions, all methods achieve relatively high success rates, with our method performing slightly better than the baselines. However, in generalization testing, baseline methods exhibit significant performance drop, while our approach maintains a high success rate, showcasing the superior adaptability of *Stem-OB* to complex and noisy real-world visual disturbances.

The experiment results demonstrate that previous visual augmentation methods, such as SRM and Mix-Spectrum, struggle to generalize in real-world scenarios, which could be attribute to the complexity of real-world environments. Real images contain more redundant information, complicating the frequency domain and potentially leading to the failure of these augmentation methods. The light disturbance introduced in our experiments is a typical example, where wide-range but low-intensity noise is added to the images. Our approach effectively handles real-world noise by extracting high-level structures from the appearances, resulting in better generalization. Even in challenging scenarios like D2B, where baseline methods mostly fail, our method maintains a high success rate.

Interestingly, DP without any image modification (Org) outperforms the baselines in some tasks, such as OD and ToF, suggesting that DP has inherent generalization capabilities. However, this generalizability is inconsistent across tasks. For instance, in C2P, we observe that object appearance (cup and plate) had little impact on DP's performance, while the tablecloth pattern significantly affects it. In D2B, the duck's appearance is critical, whereas the table cloth variation is more influential. In contrast, our method exhibits consistent generalization across diverse scenarios.

## 6.4 SIMULATION EXPERIMENTS

**Sapien 3** The evaluation results on Sapien 3 are presented on the left side of Tab. 3. The complexity of the tasks poses challenges for DP in generalizing across various conditions. However, our method achieves a higher success rate than the baseline methods on most of the tasks. In *PushCube*, both Mix-Spectrum and our method perform well, but our approach is more robust and reach nearly 100% success. These results demonstrate that in more photorealistic simulation environments, our method generalizes more effectively across diverse tasks and conditions.

**MimicGen.** The evaluation results on the MimicGen benchmark are presented on the right side of Tab. 3. At first glance, the results seem unanticipated, as *Stem-OB* does not perform best in most settings. This can be attributed to the fact that MimicGen environments are less photorealistic, with

Table 4: **Ablation study on diffusion steps.** On the left side of the table, we increase the inversion step with fixed total number of steps, to intensify the effect of the diffusion process. Additionally, we compare the performance of the model with fixed ratio of steps, where the intensity of inversion is approximately the same. The results are reported as mean over 21 kinds of settings. The best performance is highlighted in orange and the second best is highlighted in pink.

| Task \ | Fixed Total Number of Steps (50) | | | | | | Fixed Ratio of Steps (30%) | | |
|---|---|---|---|---|---|---|---|---|---|
| Settings | 0/50 | 5/50 | 10/50 | 15/50 | 20/50 | 25/50 | 9/30 | 15/50 | 30/100 |
| PushCube | 61.8 | 97.1 | 91.1 | 99.1 | 98.5 | 96.3 | 98.1 | 99.1 | 98.3 |
| PickCube | 8.6 | 12.0 | 28.0 | 25.1 | 12.7 | 8.29 | 23.7 | 25.1 | 25.6 |
| StackCube | 4.3 | 20.7 | 27.2 | 19.6 | 11.6 | 2.0 | 19.5 | 19.6 | 20.2 |

nearly texture-free and low-resolution images. Consequently, diffusion inversion processing complicates the observations, limiting DP from fully leveraging the advantages of *Stem-OB*. It is important to emphasize that our method is primarily designed for real-world scenarios, where diverse noise is inevitable, in contrast to the controlled MimicGen environment.

## 6.5 ABLATION

In this section, we test with several design choices of *Stem-OB*, providing a better understanding of their impact on the final performance. We consider the following choices: the number of inversion steps with fixed total steps, the number of total steps with the constant proportion of inversion steps to the total steps, and the selection of inversion methods (DDPM or DDIM inversion). The experiments are conducted on SAPIEN 3. We choose *PushCube*, *PickCube* and *StackCube*, since they have a more significant performance variance in the main experiments.

**Number of Inversion Steps.** We compare the performance of *Stem-OB* with varying numbers of inversion steps, fixing the total steps to 50 and adjusting the inversion steps from 5 to 25 in increments of 5. The results, shown in Tab. 4, indicate that performance generally increases up to 15 inversion steps before declining. This can be attributed to insufficient removal of low-level appearance features with too few inversion steps, while excessive steps eliminate high-level structural information, hindering task performance. Optimal performance is observed around 10 to 15 inversion steps, depending on the complexity of the tasks.

**Number of Total Steps.** With the proportion of inversion steps to total steps fixed, we varied the total number of steps to 30, 50, and 100. The results, shown on the right side of Tab. 4, indicate that performance remains consistent regardless of the total number of steps. This suggests that the proportion of inversion steps is more critical to performance than the total number of steps.

**Diffusion Inversion Methods.** We compare the performance of two inversion methods: DDPM inversion and DDIM inversion, with results presented in Tab. 5. For DDIM inversion, we use 5/50 inversion steps instead of 15/50 due to differences in noise scheduling compared to DDPM inversion. The results show that DDPM inversion outperforms DDIM inversion on the benchmark datasets, supporting our choice of DDPM inversion as the primary diffusion inversion method.

| Task | DDPM | DDIM |
|---|---|---|
| PushCube | **99.1** | 81.3 |
| PickCube | **25.1** | 5.4 |

Table 5: **Ablation study on different inversion methods.** We compare the performance of DDPM and DDIM on the tasks of *PushCube* and *PickCube*.

## 7 CONCLUSION

In this work, we propose *Stem-OB*, a straightforward preprocessing method for visual IL. By inverting the observation in the diffusion latent space for certain steps, we effectively converge different observation variations to the node it stems from, Making it invariant to unseen low-level appearance changes in the observation space. Though we only test our method on the diffusion policy method, our method is general and compatible with any visual IL baselines in theory, and the plug-and-play nature of our method makes it easy to integrate. We plan to test our method with other visual IL baselines in simulation and real tasks in the future.

**Reproducibility:** The main algorithm of our method is simple as applying the open-sourced DDPM inversion method on the dataset before training. We've provided the code for our method in the supplementary material.

ACKNOWLEDGEMENT

This work is supported by Tsinghua University Dushi Program.

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

## A    APPENDIX

### A.1    DERIVATION OF DDPM INVERSION ATTRIBUTE LOSS

The derivation of DDPM inversion is straightforward by simply letting $\sigma = \sqrt{1 - \bar{\alpha}_t}$ in Eq. (7).

### A.2    DERIVATION OF DDIM INVERSION ATTRIBUTE LOSS

We first rewrite Eq. (5) as a linear combination of $\boldsymbol{x}_0$ and a noise variable. To do so, we expand the recursive equation in Eq. (5). Note that we assume that $\boldsymbol{\epsilon}_{\boldsymbol{\theta}}(\boldsymbol{x}_{t-1}, t, \mathcal{C})$ is a Gaussian distribution with mean $\boldsymbol{\mu}_{\boldsymbol{t}}$ and use $\boldsymbol{\epsilon}_t$ for simplicity in the following derivation.

$$
\begin{aligned}
\boldsymbol{x}_t &= \sqrt{\frac{\bar{\alpha}_t}{\bar{\alpha}_{t-1}}} \boldsymbol{x}_{t-1} + \left( \sqrt{\frac{1}{\bar{\alpha}_t} - 1} - \sqrt{\frac{1}{\bar{\alpha}_{t-1}} - 1} \right) \boldsymbol{\epsilon}_t \\
&= \sqrt{\frac{\bar{\alpha}_t}{\bar{\alpha}_{t-1}}} \left( \sqrt{\frac{\bar{\alpha}_{t-1}}{\bar{\alpha}_{t-2}}} \boldsymbol{x}_{t-2} + \left( \sqrt{\frac{1}{\bar{\alpha}_{t-1}} - 1} - \sqrt{\frac{1}{\bar{\alpha}_{t-2}} - 1} \right) \boldsymbol{\epsilon}_{t-1} \right) + \left( \sqrt{\frac{1}{\bar{\alpha}_t} - 1} - \sqrt{\frac{1}{\bar{\alpha}_{t-1}} - 1} \right) \boldsymbol{\epsilon}_t \\
&= \sqrt{\frac{\bar{\alpha}_t}{\bar{\alpha}_{t-2}}} \left( \sqrt{\frac{\bar{\alpha}_{t-2}}{\bar{\alpha}_{t-3}}} \boldsymbol{x}_{t-3} + \left( \sqrt{\frac{1}{\bar{\alpha}_{t-2}} - 1} - \sqrt{\frac{1}{\bar{\alpha}_{t-3}} - 1} \right) \boldsymbol{\epsilon}_{t-2} \right) \\
&\qquad + \sqrt{\frac{\bar{\alpha}_t}{\bar{\alpha}_{t-1}}} \left( \sqrt{\frac{1}{\bar{\alpha}_{t-1}} - 1} - \sqrt{\frac{1}{\bar{\alpha}_{t-2}} - 1} \right) \boldsymbol{\epsilon}_{t-1} + \left( \sqrt{\frac{1}{\bar{\alpha}_t} - 1} - \sqrt{\frac{1}{\bar{\alpha}_{t-1}} - 1} \right) \boldsymbol{\epsilon}_t \\
&\;\; \dots \\
&= \sqrt{\bar{\alpha}_t} \boldsymbol{x}_0 + \sum_{i=1}^{t} \sqrt{\frac{\bar{\alpha}_t}{\bar{\alpha}_i}} \left( \sqrt{\frac{1}{\bar{\alpha}_i} - 1} - \sqrt{\frac{1}{\bar{\alpha}_{i-1}} - 1} \right) \boldsymbol{\epsilon}_i \\
&= \sqrt{\bar{\alpha}_t} \boldsymbol{x}_0 + \sum_{i=1}^{t} \sqrt{\frac{\bar{\alpha}_t}{\bar{\alpha}_i}} \left( \sqrt{\frac{1}{\bar{\alpha}_i} - 1} - \sqrt{\frac{1}{\bar{\alpha}_{i-1}} - 1} \right) \boldsymbol{\mu}_{\boldsymbol{t}} + \sqrt{\sum_{i=1}^{t} \frac{\bar{\alpha}_t}{\bar{\alpha}_i} \left( \sqrt{\frac{1}{\bar{\alpha}_i} - 1} - \sqrt{\frac{1}{\bar{\alpha}_{i-1}} - 1} \right)^2} \boldsymbol{\epsilon}
\end{aligned}
\tag{12}
$$

where the last equality holds considering the property of sum of gaussian distributions, i.e., $\mathcal{N}(\boldsymbol{\mu}_1, \sigma_1^2 \boldsymbol{I}) + \mathcal{N}(\boldsymbol{\mu}_2, \sigma_2^2 \boldsymbol{I}) \sim \mathcal{N}(\boldsymbol{\mu}_1 + \boldsymbol{\mu}_2, (\sigma_1^2 + \sigma_2^2)\boldsymbol{I})$. And $\sum_{i=1}^{t} \sqrt{\frac{\bar{\alpha}_t}{\bar{\alpha}_i}} \left( \sqrt{\frac{1}{\bar{\alpha}_i} - 1} - \sqrt{\frac{1}{\bar{\alpha}_{i-1}} - 1} \right) \boldsymbol{\mu}_{\boldsymbol{t}}$ represents the mean shift resulting from the mean bias of model $\boldsymbol{\epsilon}_{\boldsymbol{\theta}}$. For two similar images, the biases are approximately equal and thus cancel out when substituted into Eq. (7). Consequently, we can approximately derive the attribute loss for DDIM inversion.

$$
\begin{aligned}
loss_{DDIM}(\boldsymbol{x}_0, \boldsymbol{y}_0, t) &\approx \frac{1}{2}\left[ 1 - \operatorname{erf}\left( \frac{\|\sqrt{\bar{\alpha}_t}(\boldsymbol{y}_0 - \boldsymbol{x}_0)\|}{2\sqrt{2\sum_{i=1}^{t} \frac{\bar{\alpha}_t}{\bar{\alpha}_i} \left( \sqrt{\frac{1}{\bar{\alpha}_i} - 1} - \sqrt{\frac{1}{\bar{\alpha}_{i-1}} - 1} \right)^2}} \right) \right] \\
&= \frac{1}{2}\left[ 1 - \operatorname{erf}\left( \frac{\|(\boldsymbol{y}_0 - \boldsymbol{x}_0)\|}{2\sqrt{2\sum_{i=1}^{t} \frac{1}{\bar{\alpha}_i} \left( \sqrt{\frac{1}{\bar{\alpha}_i} - 1} - \sqrt{\frac{1}{\bar{\alpha}_{i-1}} - 1} \right)^2}} \right) \right]
\end{aligned}
\tag{13}
$$

## B    TRAINING DETAILS

The hyperparameters for Diffusion Policy across all experiments are listed in Tab. 6. We use DDPM inversion in all the basic experiments, with the specific inversion steps for each experiment detailed in Tab. 7.

| Hyperparameter | value | Hyperparameter | value |
|---|---|---|---|
| epoch_every_n_steps | 100 | ddim_num_train_timesteps | 100 |
| seq_length | 15 | ddim_num_inference_timesteps | 10 |
| frame_stack | 2 | ddim_beta_schedule | squaredcos_cap_v2 |
| batch_size | 128 | ddim_clip_sample | TRUE |
| num_epochs | 1500 | ddim_set_alpha_to_one | TRUE |
| learning_rate_initial | 0.0001 | ddim_steps_offset | 0 |
| learning_rate_decay_factor | 0.1 | ddim_prediction_type | epsilon |
| regularization | L2/0.0 | VisualCore_feature_dimension | 64 |
| observation_horizon | 2 | VisualCore_backbone_class | ResNet18Conv |
| action_horizon | 8 | VisualCore_pool_class | SpatialSoftmax |
| prediction_horizon | 16 | VisualCore_pool_num_kp | 32 |
| unet_diffusion_embed_dim | 256 | VisualCore_pool_temperature | FALSE |
| unet_down_dims | [256, 512,1024] | VisualCore_pool_noise_std | 1 |
| unet_kernel_size | 4 | obs_randomizer_class | CropRandomizer |
| unet_n_groups | 8 | num_crops | 1 |
| ema_power | 0.75 | | |

Table 6: **Hyperparameters used for Diffusion Policy.** We use the same hyperparameters of diffusion policy for all the experiments.

| Task | Step | Task | Step | Task | Step |
|---|---|---|---|---|---|
| Cup2Plate | 15/50 | PushCube | 15/50 | MugCleanup | 10/50 |
| Duck2Bowl | 15/50 | PegInsertionSide | 15/50 | Threading | 5/50 |
| OpeningDrawer | 15/50 | PickCube | 15/50 | ThreePieceAssembly | 5/50 |
| Turn on Faucet | 15/50 | StackCube | 10/50 | Coffee | 5/50 |

Table 7: **Tasks and inversion steps used for training the Diffusion Policy.** We all use DDPM as the inversion method.

## C EXPERIMENT DETAILS

### C.1 REALWORLD DETAILS

Fig. 6 shows the camera views of the real-world environments. Each task is represented by two rows, corresponding to different camera angles. The columns represent task settings, with the first being the default. As lighting conditions vary across tasks, images are grouped in pairs: the odd columns are under warm light and the even columns are under cold light. The low-level object appearance changes along each row.

### C.2 SAPIEN ENVIRONMENTS SETTINGS

Each task includes 21 distinct testing environments, as shown in Fig. 7. We alternate between seven unique tablecloth textures and three lighting conditions: white, yellow, and red light.

### C.3 MIMICGEN ENVIRONMENTS SETTINGS

Fig. 9 demonstrates the four MimicGen environments used in the experiments, and each row in Fig. 8 represents a task. In *MugCleanup*, the robot arm opens a drawer, retrieves a mug, and places it back. *Threading* involves the precise insertion of a stick into a hole. *ThreePieceAssembly* requires collecting and assembling three components in a specific order. In *Coffee*, the robot opens the lid of the coffee machine and places a cup inside. Eight different table textures are used across all tasks (first eight columns). Additionally, *MugCleanup*, *Threading*, and *ThreePieceAssembly* employ alternating object texture, as indicated in the last column, creating distinct testing environments.

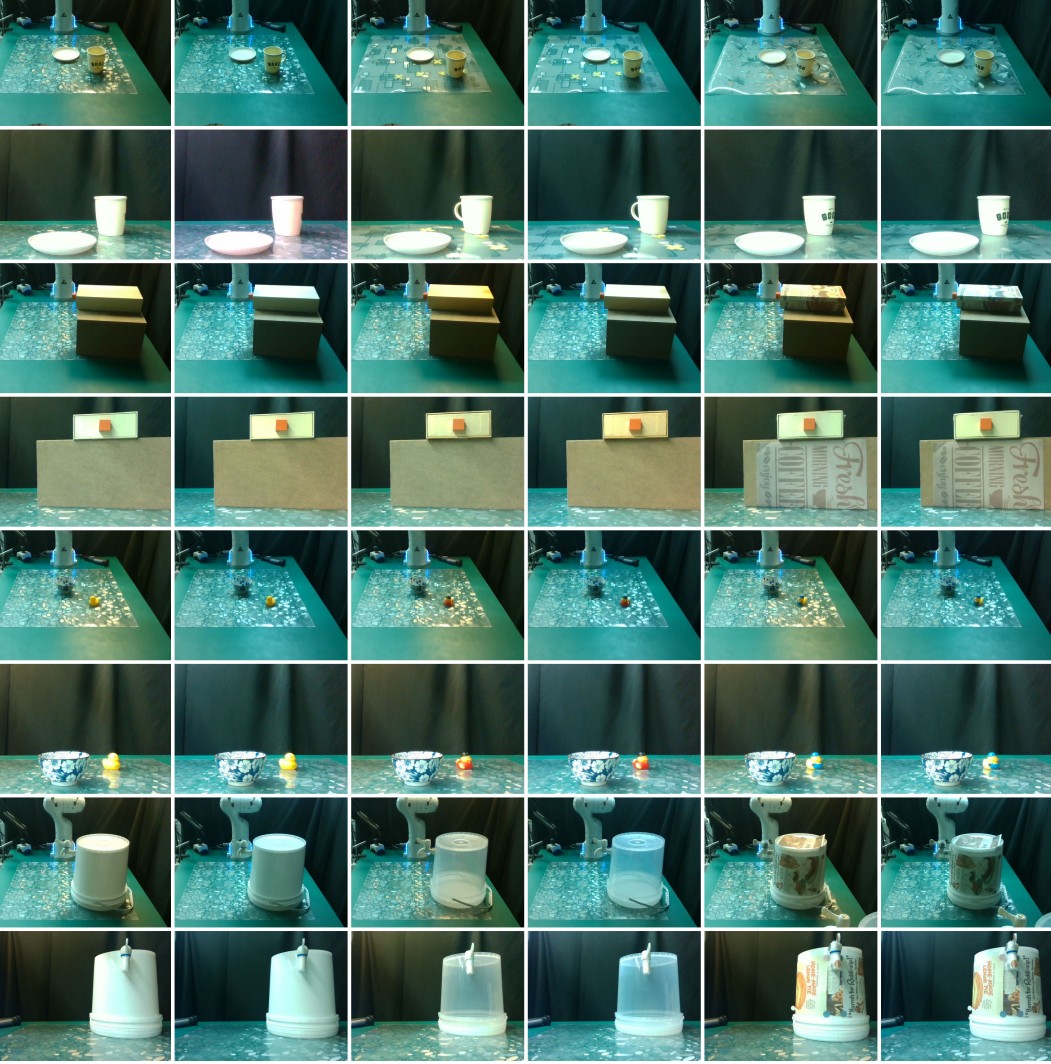

Figure 6: **Realworld Camera Views.** Each task is represented by two rows, with each representing a different camera view angle. For each task, we perform experiments on three instances with different object appearances, and two light conditions, as displayed along the columns. The first column represents the training setup.

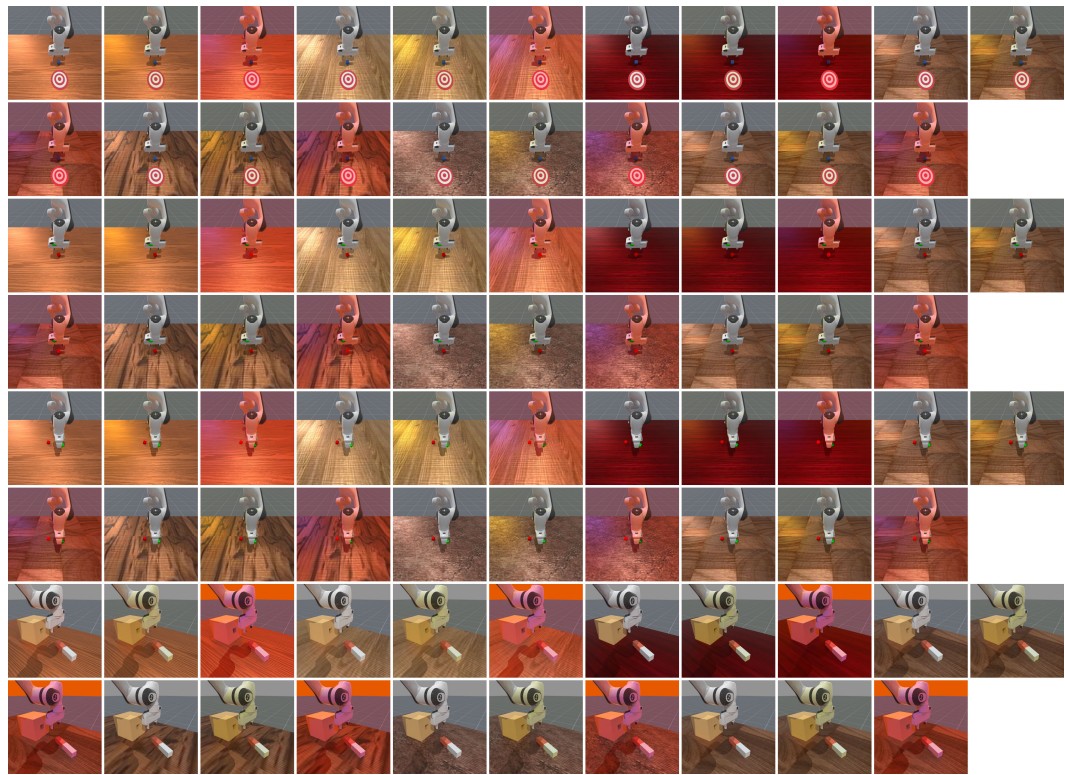

Figure 7: **Sapien Testing Settings.** Each task consists of 21 distinct settings, arranged in two rows. The images are grouped in threes, with each group sharing the same tablecloth texture but differing in lighting conditions: white, yellow, and red light. A total of seven unique tablecloth textures are used. The training environment is the first setting in the first row.

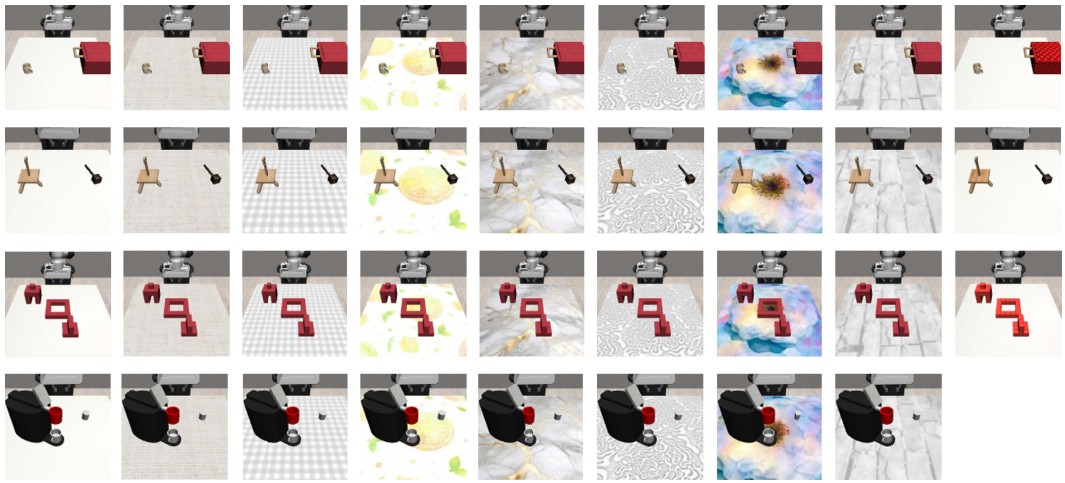

Figure 8: **MimicGen Testing Settings.** Each row represents a specific task, with eight different table textures across the columns. The training environment is the first column. Additionally, *MugCleanup*, *Threading*, and *ThreePieceAssembly* employ alternate object textures in the last column.

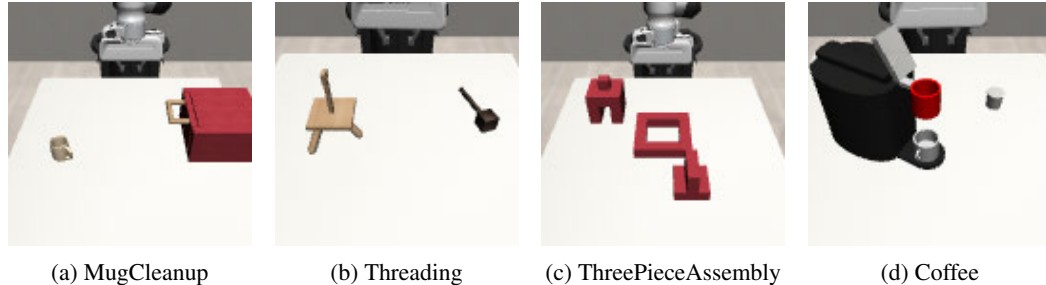

(a) MugCleanup     (b) Threading     (c) ThreePieceAssembly     (d) Coffee

Figure 9: **MimicGen environments.** These tasks are based on MimicGen benchmark. (a) The agent must open the drawer, pick up the mug, and place it into the drawer. (b) This task requires the agent to thread a string through a hole. (c) The agent must assemble three pieces together. (d) The agent is required to open the lid, and then place the coffee cup inside it.

## C.4 DETAILS OF EXPERIMENT RESULTS

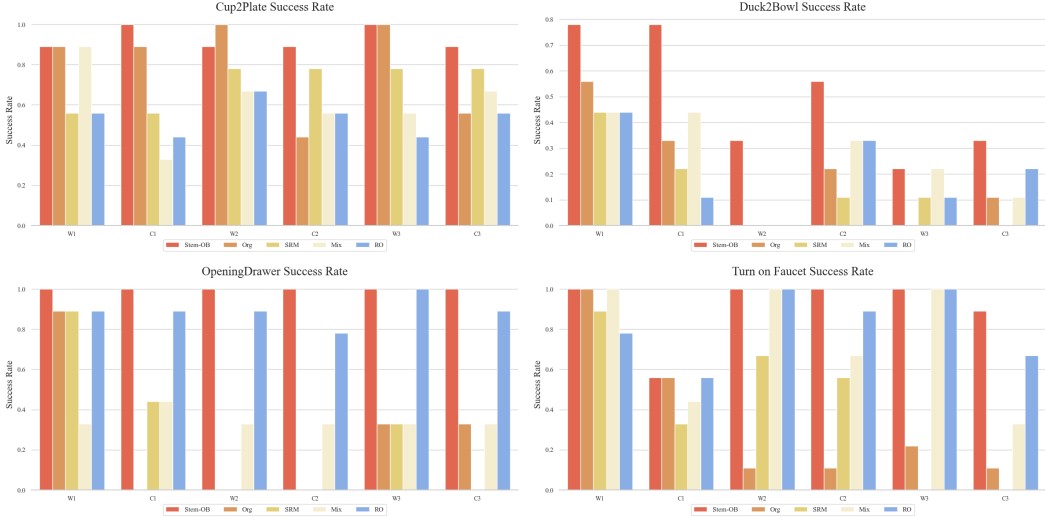

Figure 10: **Realworld Success Rate.** This figure presents the success rates for real-world tasks. The groups are labeled as W1, C1 through W3, C3. W1 to W3 are conducted under the same warm lighting condition as the training set with different object appearances. C1 to C3 are performed under identical cold lighting conditions, with varying object appearances. The order of the groups is consistent to that shown in Fig. 6.

Fig. 10, Fig. 11, and Fig. 12 illustrate the success rates of *Stem-OB* and baseline methods in real-world, SAPIEN 3, and MimicGen environments, respectively. These three figures display the performance of *Stem-OB* under each setting with more details.

## C.5 DETAILS OF ILLUSTRATIVE EXPERIMENTS

*Image Distance Calculation*: We use the 2-norm to compute the image distance $D(\boldsymbol{x}, \boldsymbol{y})$ in the latent space, where a smaller distance indicates greater similarity between the two images.

*Intra-Category Distance Calculation*: the intra-category distance is calculated as the mean of the pairwise distances between images within the same category $\mathcal{I}$

$$D_{intra} = \frac{2}{N(N-1)} \sum_{i=1}^{N} \sum_{j=i+1}^{N} D(\boldsymbol{x}_i, \boldsymbol{x}_j) \tag{14}$$

where $N$ is the number of images within $\mathcal{I}$ and $\boldsymbol{x}_i$ is the $i$-th image.

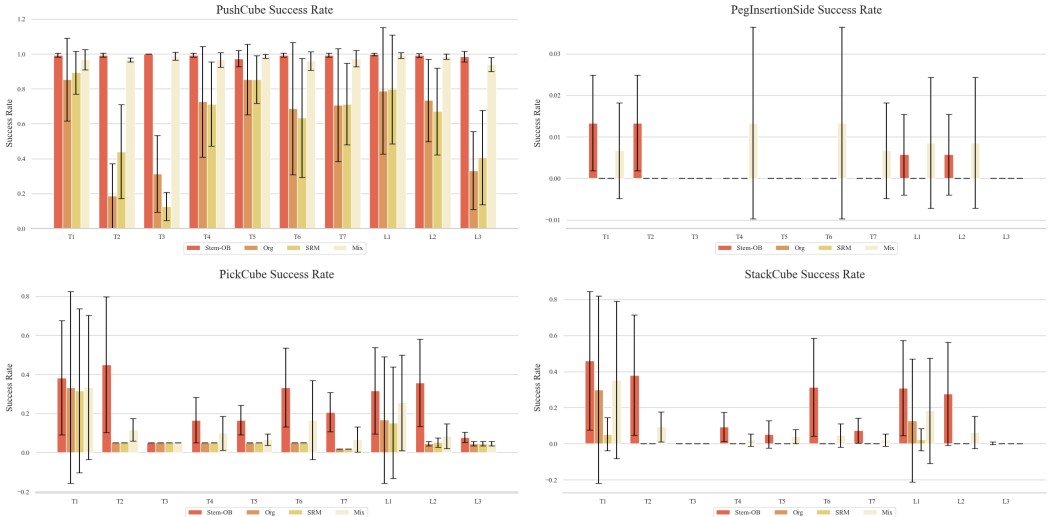

Figure 11: **Sapien Success Rate.** This figure presents the success rates for tasks of the SAPIEN 3 benchmark. The groups are labeled as T1 to T7 and L1 to L3. The settings are varied across 7 different tablecloth textures (T1 to T7) and 3 distinct lighting conditions (L1 to L3). For T1 to T7, the mean and standard deviation of the success rates are calculated over different lighting conditions with 3 lighting conditions combined for one texture. Conversely, for L1 to L3, the mean and standard deviation are computed over various tablecloth textures under a fixed lighting condition, with 7 textures evaluated for each light condition. The order of tablecloth textures and lighting conditions aligns with that presented in Fig. 7.

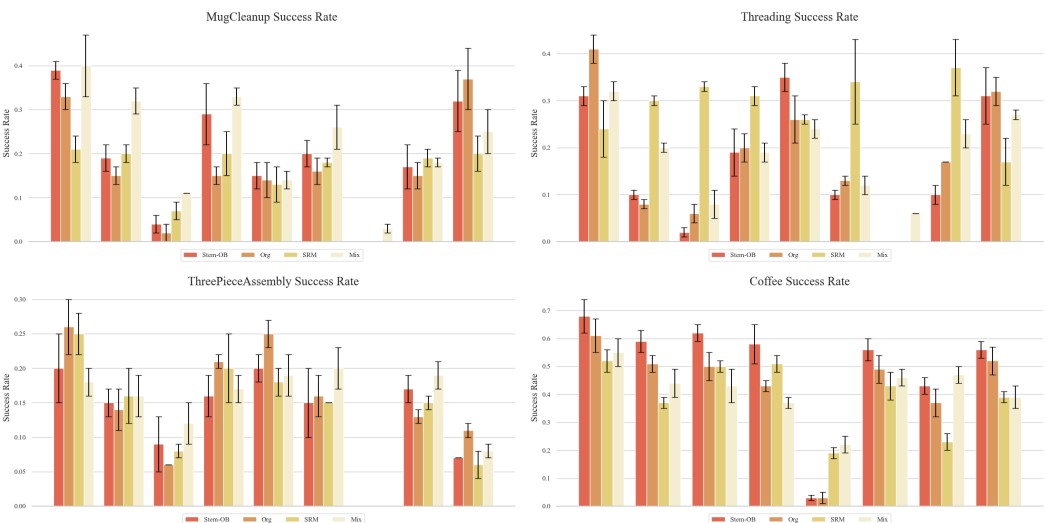

Figure 12: **MimicGen Success Rate.** This figure presents the success rates of *Stem-OB* and baselines across four distinct tasks. The mean and standard deviation are computed over 300 episodes. Each group of bars corresponds to one experimental setting. The bar order is consistent with the arrangement shown in Fig. 8.

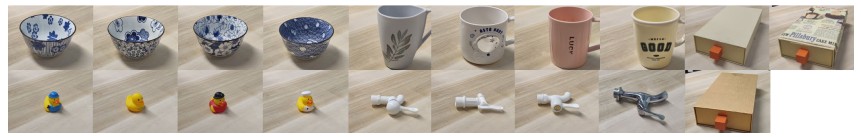

Figure 13: **Illustrative Experiment Objects.**

*Cross-Category Distance Calculation*: the cross-category distance is calculated as the mean of the pairwise distances between images from two different categories $\mathcal{I}_1$ and $\mathcal{I}_2$

$$D_{intra} = \frac{1}{MN} \sum_{i=1}^{M} \sum_{j=1}^{N} D(\boldsymbol{x}_i, \boldsymbol{y}_j) \tag{15}$$

where $M$ and $N$ are the size of $\mathcal{I}_0$ and $\mathcal{I}_1$, respectively, $\boldsymbol{x}_i \in \mathcal{I}$ and $\boldsymbol{y}_i \in \mathcal{I}_2$.

### C.6 DETAILS OF USER STUDY

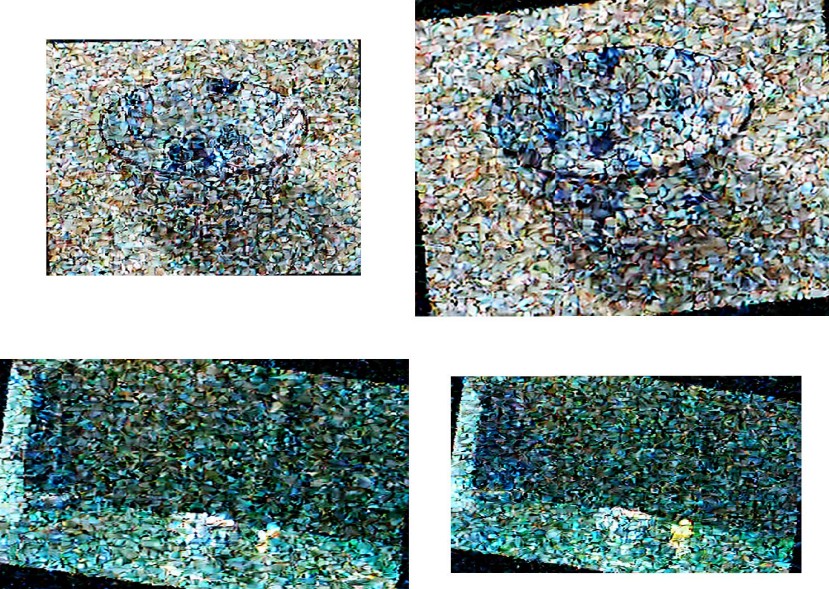

Figure 14: **Example of User Study Questions.** The upper two images represent one example from the first section, and the lower two images represent one example from the second section. Each pair is generated through a randomly selected inversion step. To prevent users from making judgments based on image size or orientation, these images have been randomly rotated, cropped, and zoomed.

We conducted a user study to evaluate the quality of the inversion results through a questionnaire-based approach. In the first section, consisting of 42 questions, participants compare two images per question and are asked to determine whether the images depict the same object, different objects of the same category, or entirely different objects. Participants are informed that the images have been randomly rotated, cropped, and zoomed, so image size or orientation should not influence their judgment of object similarity. All images are snapshots of items like bowls and faucets that are used in the real-world experiment, with each pair inverted by a randomly selected inversion step.

The second section, consisting of 56 questions, presented images from real-world tasks, as seen in Fig. 6. In this section, participants evaluate whether the images represent the same task, different scenes from the same task (with variations such as lighting or object appearance), or entirely different tasks, where the target of the task changed (for example, a cup versus a drawer). Participants are informed that scene variations could include changes in environmental conditions, such as lighting, or the appearance of objects, like a change in a tablecloth or the texture of a drawer, but the underlying task or function remains the same. To avoid bias from image size or angle, we also randomly cropped, rotated, and zoomed the images. The example of images from the two sections is shown in Fig. 14. A total of 51 valid questionnaires were collected, with the results presented in Fig. 3.

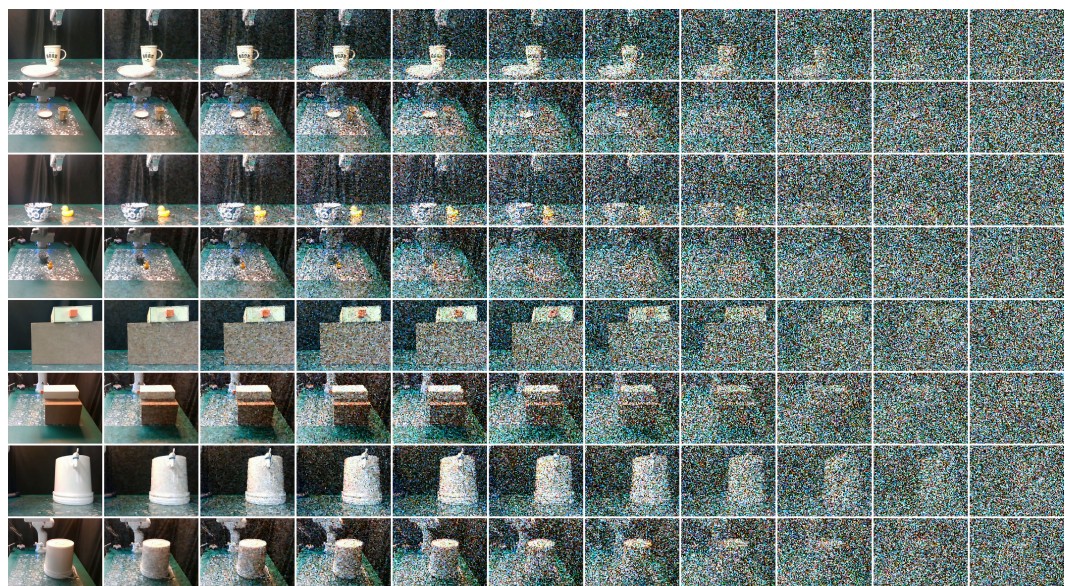

Figure 15: **Realworld Inversion Results.** Each task corresponds to two rows, with each representing a distinct camera view. The first column of each row is the original image, while the other columns are the DDPM inversion results. Starting from the second column, the diffusion steps are increased from 5 to 50, with the incremental step being 5.

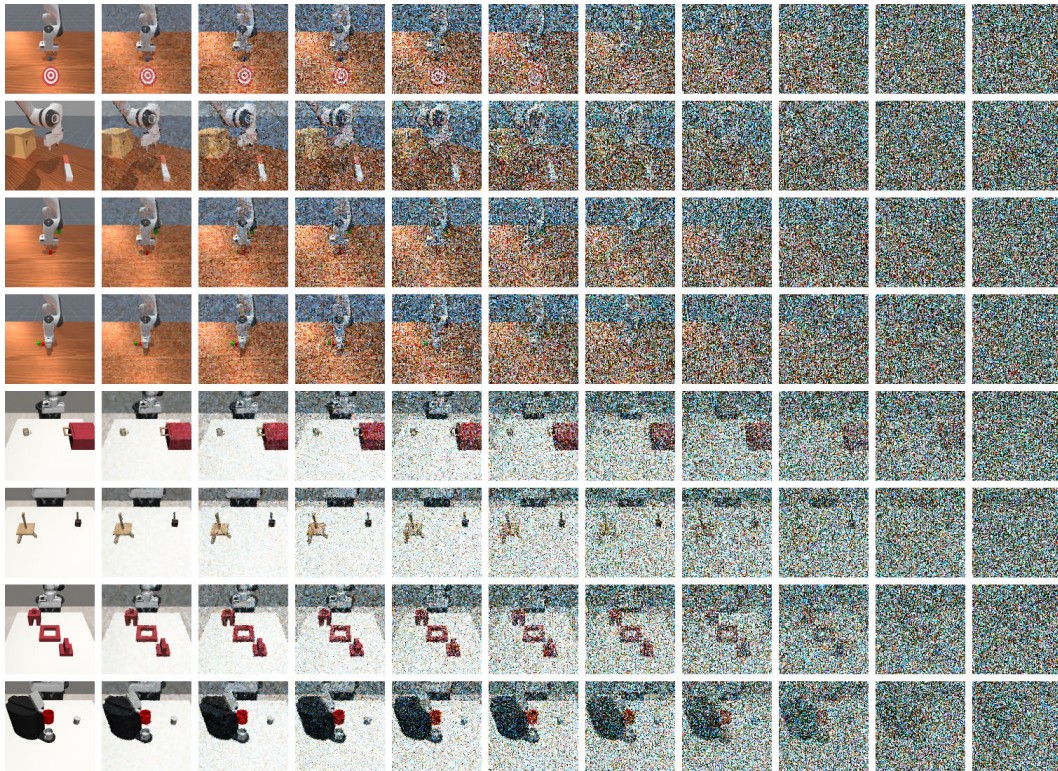

Figure 16: **Simulation Inversion Results.** The first four rows correspond to the tasks in SAPIEN 3, while the last four rows correspond to the tasks in MimicGen. Each row represents a task, with the first column displaying the original image and the subsequent columns illustrating the DDPM inversion results. Starting from the second column, the diffusion steps are increased from 5 to 50, with the incremental step being 5.

Table 8: **Success rate of SAPIEN tasks.** Train.: evaluations in the same settings as the training dataset. Gen.: evaluations under different visual perturbations for generalizability analysis. All: evaluations including both Train. and Gen. The tasks are *PushCube*, *PegInsertionSide*, *PickCube*, and *StackCube*. We report the mean and standard deviation of the success rate (%), and the best results are highlighted in **bold**. In the following tables, when "Train.", "Gen.", or "All" is mentioned, they carry the same meaning as defined above.

| Task \ | *PushCube* | | | *PegInsertionSide* | | | *PickCube* | | | *StackCube* | | |
|---|---|---|---|---|---|---|---|---|---|---|---|---|
| Algorithm | Train. | Gen. | All | Train. | Gen. | All | Train. | Gen. | All | Train. | Gen. | All |
| *Stem-OB* | **100.0** | **99.1** | **99.1** | **2.0** | 0.3 | 0.4 | 50.0 | **23.9** | **25.1** | 90.0 | **24.1** | **27.2** |
| Org | **100.0** | 59.9 | 61.8 | 0.0 | 0.0 | 0.0 | **90.0** | 4.6 | 8.6 | **90.0** | 0.0 | 4.3 |
| SRM | **100.0** | 60.6 | 62.5 | 0.0 | 0.0 | 0.0 | 80.0 | 4.8 | 8.4 | 16.0 | 0.0 | 0.7 |
| Mix | **100.0** | 97.1 | 97.2 | **2.0** | **0.5** | **0.6** | 75.0 | 9.8 | 12.9 | 84.0 | 4.4 | 8.2 |

Table 9: **Success rate of MimicGen tasks.** The tasks are *MugCleanup*, *Threading*, *ThreePieceAssembly*, and *Coffee*. We report the mean and standard deviation of the success rate (%), and the best results are highlighted in **bold**.

| Task \ | *MugCleanup* | | | *Threading* | | | *ThreePieceAssembly* | | | *Coffee* | | |
|---|---|---|---|---|---|---|---|---|---|---|---|---|
| Algorithm | Train. | Gen. | All | Train. | Gen. | All | Train. | Gen. | All | Train. | Gen. | All |
| *Stem-OB* | $38.7_{\pm 2.5}$ | $17.1_{\pm 4.4}$ | $19.5_{\pm 4.2}$ | $31.3_{\pm 2.5}$ | $14.5_{\pm 3.2}$ | $16.4_{\pm 3.1}$ | $19.7_{\pm 4.8}$ | $12.3_{\pm 2.8}$ | $13.1_{\pm 3.1}$ | $\mathbf{68.0_{\pm 5.7}}$ | $\mathbf{48.0_{\pm 3.9}}$ | $\mathbf{50.5_{\pm 4.1}}$ |
| Org | $33.0_{\pm 2.8}$ | $14.4_{\pm 3.5}$ | $16.4_{\pm 3.4}$ | $\mathbf{41.3_{\pm 3.3}}$ | $15.2_{\pm 2.4}$ | $18.1_{\pm 2.5}$ | $\mathbf{26.3_{\pm 4.5}}$ | $13.4_{\pm 1.8}$ | $\mathbf{14.8_{\pm 2.3}}$ | $61.0_{\pm 6.5}$ | $40.7_{\pm 4.1}$ | $43.3_{\pm 4.5}$ |
| SRM | $21.0_{\pm 2.9}$ | $14.7_{\pm 3.0}$ | $15.4_{\pm 3.0}$ | $24.3_{\pm 5.7}$ | $\mathbf{26.0_{\pm 4.4}}$ | $\mathbf{25.8_{\pm 4.6}}$ | $24.7_{\pm 2.9}$ | $12.3_{\pm 2.4}$ | $13.6_{\pm 2.5}$ | $52.3_{\pm 3.7}$ | $37.6_{\pm 2.7}$ | $39.4_{\pm 2.9}$ |
| Mix | $\mathbf{39.7_{\pm 6.9}}$ | $\mathbf{20.1_{\pm 3.0}}$ | $\mathbf{22.3_{\pm 3.6}}$ | $32.0_{\pm 1.6}$ | $17.3_{\pm 2.1}$ | $18.9_{\pm 2.1}$ | $17.7_{\pm 1.7}$ | $\mathbf{13.8_{\pm 2.2}}$ | $14.2_{\pm 2.2}$ | $55.3_{\pm 5.3}$ | $39.8_{\pm 3.8}$ | $41.8_{\pm 4.0}$ |

## D  VISUALIZATION

### D.1  INVERSION VISUALIZATION

Fig. 15 and Fig. 16 display the inversion results for real-world and simulation environments, respectively. In the real-world environments, each task spans two rows, corresponding to the same camera view as in Fig. 6. In the simulation environments, each row represents a task, with the first four rows showing SAPIEN 3 environments and the last four showing MimicGen environments. The first column in each row is the original image, while subsequent columns present the inverted results. From the second column onward, the inversion results are generated by DDPM inversion, with 50 total inversion steps and 5 incremental steps from 5 to 50.

## E  DETAILS OF SUCCESS RATES

We separate the results for training and test variants to provide a clearer picture of the policy's performance. For SAPIEN tasks, we test 50 episodes for each task, while for MimicGen tasks, we test 300 episodes. As is shown in the Tab. 8 and Tab. 9, *Stem-OB* performs well in the generalizing settings, while remaining competitive in the training settings.

The standard deviation of the success rates is calculated as below. For Mimicgen tasks, consider one single task. There are $n$ kinds of visual generalization variations $v_i, i = 1, ..., n$. for each setting $v_i$, we divide the 300 episodes into 3 groups and calculate the success rate mean $\mu_i$ and standard deviation $\sigma_i$ between each group. The overall mean for each task is the average of the individual means $\mu = \frac{1}{n}\Sigma_{i=1}^{n}\mu_i$, and the overall standard deviation is computed as the square root average across all settings $\sigma = \sqrt{\frac{1}{n}\Sigma_{i=1}^{n}\sigma_i^2}$.

For the SAPIEN tasks, due to time constraints, we are only able to test 50 episodes for each setting in each task. As a result, we do not have enough data to group the results and compute the mean and standard deviation for each setting. We can use hypothesis testing to demonstrate that Stem-OB significantly outperforms all the baselines.

Suppose the success rate in the given setting for a specific task is drawn from unknown geometric distributions with unknown parameters $p$. We have observed the success rates from each group over $n$ samples. Let $p_1$ represent Stem-OB's success rate, and $p_2$ represent the success rate of a baseline. We aim to test the null hypothesis $H_0 : p_1 \leq p_2$, which is a one-tailed test. The standard error of the difference between two independent proportions is given by: $SE(D) = \sqrt{\frac{\hat{p_1}(1-\hat{p_1})}{n} + \frac{\hat{p_2}(1-\hat{p_2})}{n}}$,

Table 10: **Significant count.** The data is presented as $n/m$, where $n$ represents the number of instances where $z > 1.645$, which means Stem-OB significantly outperform the corresponding baseline, and $m$ represents the number of instances where $z < -1.645$, indicating the contrast.

| Task | *PushCube* | *PegInsertionSide* | *PickCube* | *StackCube* |
|------|------------|--------------------|------------|-------------|
| **Org** | 14/0 | 0/0 | 11/1 | 11/0 |
| **SRM** | 16/0 | 0/0 | 11/1 | 12/0 |
| **Mix** | 4/0 | 0/0 | 9/1 | 11/0 |

Table 11: **Success rate of Additional MimicGen tasks.** The tasks are *Square*, *StackThree*, *Stack*, and *CoffeePreparation*. We report the mean and standard deviation of the success rate (%), and the best results are highlighted in **bold**.

| Task \ | *Square* | | | *StackThree* | | | *Stack* | | | *CoffeePreparation* | | |
|--------|----------|----------|----------|--------------|----------|----------|---------|----------|----------|---------------------|----------|----------|
| Algorithm | Train. | Gen. | All | Train. | Gen. | All | Train. | Gen. | All | Train. | Gen. | All |
| *Stem-OB* | 50.7$_{\pm1.7}$ | 28.3$_{\pm3.5}$ | 31.4$_{\pm3.3}$ | 41.3$_{\pm0.9}$ | **27.6$_{\pm1.8}$** | **29.3$_{\pm1.7}$** | 85.7$_{\pm0.9}$ | 70.7$_{\pm2.8}$ | 72.6$_{\pm2.6}$ | 14.7$_{\pm2.1}$ | **13.2$_{\pm1.8}$** | **13.4$_{\pm1.8}$** |
| Org | 53.3$_{\pm5.2}$ | 22.0$_{\pm3.1}$ | 25.9$_{\pm3.4}$ | **44.0$_{\pm5.1}$** | 25.7$_{\pm1.6}$ | 28.0$_{\pm3.5}$ | **87.3$_{\pm2.1}$** | **74.5$_{\pm2.5}$** | **76.1$_{\pm2.5}$** | **18.7$_{\pm5.0}$** | 10.1$_{\pm1.6}$ | 11.1$_{\pm2.3}$ |
| SRM | 51.7$_{\pm0.5}$ | 22.4$_{\pm3.8}$ | 26.1$_{\pm3.6}$ | 28.0$_{\pm0.0}$ | 20.4$_{\pm1.8}$ | 21.3$_{\pm1.7}$ | 75.3$_{\pm3.4}$ | 58.7$_{\pm4.0}$ | 60.8$_{\pm3.8}$ | 15.0$_{\pm2.2}$ | 9.5$_{\pm2.9}$ | 10.2$_{\pm2.8}$ |
| Mix | **53.7$_{\pm5.6}$** | **37.5$_{\pm4.7}$** | **39.5$_{\pm4.8}$** | 35.7$_{\pm2.5}$ | 18.3$_{\pm3.2}$ | 20.5$_{\pm3.1}$ | 79.7$_{\pm3.1}$ | 66.8$_{\pm3.8}$ | 68.4$_{\pm3.7}$ | 17.7$_{\pm2.5}$ | 11.9$_{\pm2.8}$ | 12.6$_{\pm2.8}$ |

where $D = \hat{p}_1 - \hat{p}_2$. So The test statistic for this hypothesis test is calculated as the difference divided by the standard error $z = \frac{D}{SE(D)}$. To perform the hypothesis test at a 95% confidence level, the critical value for a one-tailed test is 1.645. For the null hypothesis to be rejected (i.e., to conclude that $p_1 > p_2$), the test statistic must exceed this critical value.

On the SAPIEN tasks, we calculate the z-value of Stem-OB for each setting against all three baselines. The number in Tab. 10 is presented as $n/m$, where $n$ represents the number of instances where $z > 1.645$, which means Stem-OB significantly outperform the corresponding baseline, and $m$ represents the number of instances where $z < -1.645$, indicating Stem-OB significantly underperform the corresponding baseline.

Tab. 10 shows that Stem-OB significantly outperforms the baselines in most settings across these tasks, with very few or no instances of underperformance. Therefore, we can conclude that Stem-OB demonstrates superior generalization ability compared to the baselines, with statistical significance.

# F ADDITIONAL MIMICGEN EXPERIMENTS

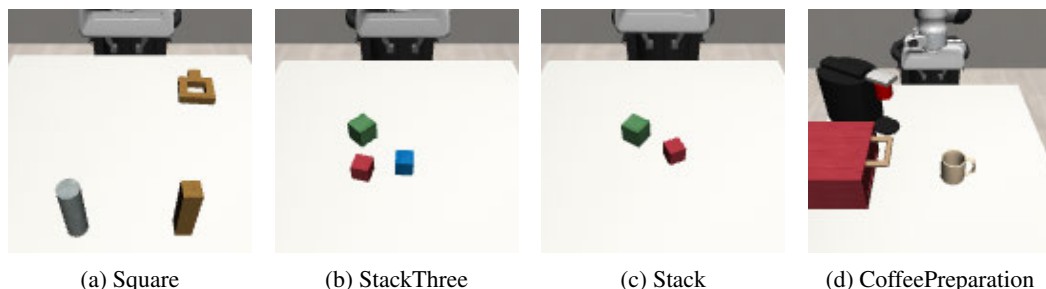

(a) Square      (b) StackThree      (c) Stack      (d) CoffeePreparation

Figure 17: **Additional MimicGen environments.** These tasks are based on MimicGen benchmark. (a) The agent must pick up square nuts and place them onto the correct pole. (b) The agent is required to stack three cubes in the correct order. (c) The task also needs stacking but only with two cubes. (d) The agent need to place a mug on the machine, open the lid, open the drawer to get coffee pod and insert it into the machine.

To further strengthen our evaluation, we add experiments for additional MimicGen tasks. We report results for an additional 4 tasks, including **Square**, **StackThree**, **Stack**, and **CoffeePreparation**. Fig. 17 shows the images for each task. In **Square**, the robot arm must pick up square nuts and place them onto the correct pole. In **StackThree**, the robot arm is asked to stack three cubes in the correct order. **Stack** is similar to **StackThree**, but with only two cubes. **CoffeePreparation** is a long-

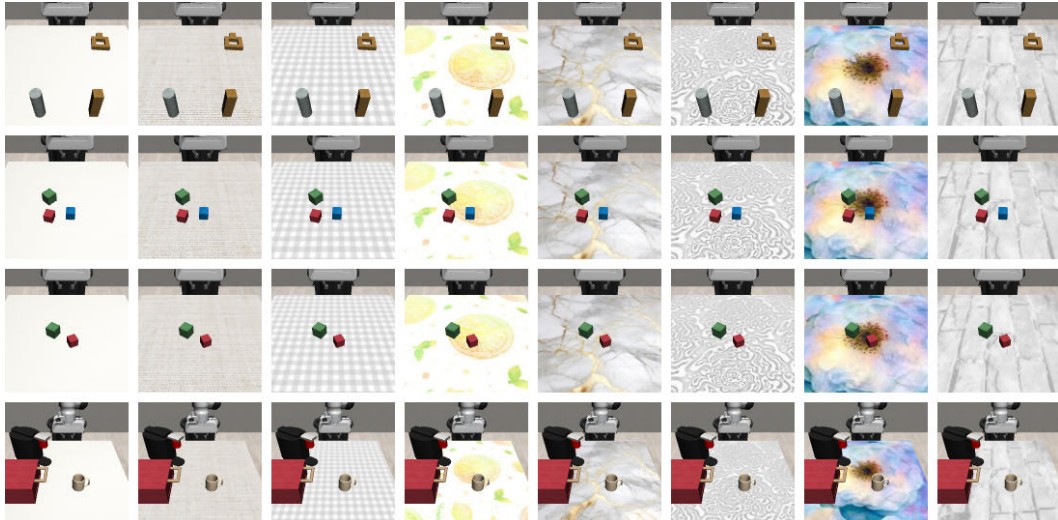

Figure 18: **Additional MimicGen Testing Settings.** Each row represents a specific task, with eight different table textures across the columns. The training environment is the first column.

Table 12: **Success rate of RO in SAPIEN.** The tasks are *PushCube*, *PickCube*, and *StackCube*. We report the mean and standard deviation of the success rate (%), and the best results are highlighted in **bold**. We constructed a simulation dataset by overlaying images from the MimicGen dataset to adapt the RO algorithm to simulation datasets.

| Task \ | *PushCube* | | | *PickCube* | | | *StackCube* | | |
|---|---|---|---|---|---|---|---|---|---|
| Algorithm | Train. | Gen. | All | Train. | Gen. | All | Train. | Gen. | All |
| *Stem-OB* | **100.0** | **99.1** | **99.1** | 50.0 | **23.9** | 25.1 | **90.0** | **24.1** | **27.2** |
| RO | **100.0** | 92.1 | 92.5 | **86.0** | 22.5 | **25.5** | 60.0 | 7.2 | 9.7 |

horizon task where the robot arm must prepare a cup of coffee by placing a mug under the coffee machine, opening the lid of the coffee machine, then retrieving the coffee pod from a drawer and finally inserting it in the machine. The test settings are shown in Fig. 18. Tab. 11 presents the success rates for these tasks. *Stem-OB* achieves competitive performance across all tasks, demonstrating its generalization capabilities.

# G    RANDOM OVERLAY BASELINE IN SAPIEN ENVIRONMENTS

The original random overlay baseline uses the Places365 dataset, which contains real-world images inconsistent with the simulation environments, thus we didn't take it into account in simulation tasks. To adapt it for simulation, we constructed a simulation dataset by overlaying images from the MimicGen dataset. Specifically, we randomly selected one of the seven MimicGen environments and picked an arbitrary image from a random trajectory to overlay onto the original image. Using this adapted RO dataset, we evaluated **RO** on *PushCube*, *PickCube* and *StackCube*. The results show that *Stem-OB* still outperforms **RO** in most tasks as shown in Tab. 12.

# H    MORE GENERALIZATION EXPERIMENTS

We introduced a new type of visual variance in the SAPIEN environment by adding normal maps to the tabletop and combining this perturbation with other variations, such as lighting conditions and table textures. Fig. 19 provides a visualization of these new perturbations. We then evaluated *Stem-OB* and the baselines on the *PushCube* and *PickCube* tasks. The results are shown in Tab. 13. *Stem-OB* consistently outperforms the baselines, even when encountering previously unseen perturbations.

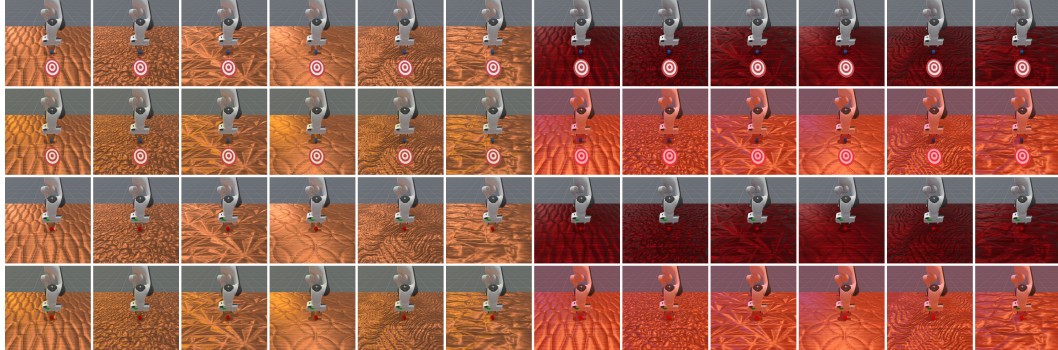

Figure 19: **Sapien Testing Settings.** Each task consists of 24 distinct settings, arranged in two rows. The first row of each task contains 12 settings, with the left half showing 6 *normal only* perturbation and the right half showing the *normal + texture* perturbation. The second row of each task represents 12 *normal+lighting* perturbation settings, with 2 lights combined with 6 normal maps.

Table 13: **Evaluations in SAPIEN with normal map perturbation.** normal only: evaluations with normal map perturbation only. normal+texture: evaluations with normal map and texture perturbation. normal+light: evaluations with normal map and light perturbation. We report the mean of the success rate (%), and the best results are highlighted in **bold**.

| Task \ | *PushCube* | | | *PickCube* | | |
|---|---|---|---|---|---|---|
| Algorithm | normal only | normal+texture | normal+light | normal only | normal+texture | normal+light |
| *Stem-OB* | 98.0 | **99.3** | 98.7 | **30.0** | **2.3** | **17.7** |
| Org | 74.7 | 39.0 | 58.5 | 2.0 | 2.0 | 2.0 |
| SRM | 86.0 | 8.3 | 75.8 | 2.3 | **2.3** | 2.2 |
| Mix | **99.0** | 97.3 | **99.0** | 22.3 | 2.0 | 8.5 |

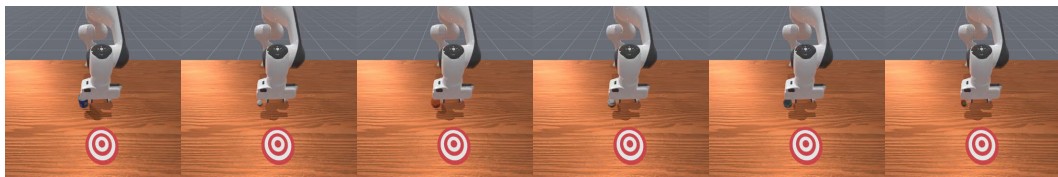

Figure 20: **Cross-category PushCube Testing Settings.** Each picture represents a test setting a random selected object from YCB object sets. The order of the setting is the same as that in the Tab. 14.

Table 14: **Success rate of Cross-category settings in *PushCube*.** The original cube object is replaced with 6 different and random selected object from YCB object sets, such as a golf ball. We use DDPM as the inversion model in this experiment. We report the mean of the success rate (%), and the best results are highlighted in **bold**.

| Settings | 15/50 | 20/50 | 25/50 | org | srm | mix |
|---|---|---|---|---|---|---|
| chef_can | 0.30 | **0.34** | **0.34** | 0.30 | 0.26 | 0.28 |
| golf | **0.30** | 0.26 | **0.30** | 0.24 | 0.24 | 0.26 |
| orange | 0.34 | 0.42 | **0.44** | 0.32 | 0.34 | 0.42 |
| soccer | 0.30 | **0.40** | **0.40** | 0.28 | 0.30 | 0.34 |
| fish_can | 0.24 | **0.34** | **0.34** | 0.20 | 0.28 | **0.34** |
| strawberry | 0.20 | 0.24 | **0.28** | 0.16 | **0.28** | 0.26 |

Table 15: **Success rate of SSL in SAPIEN.** The tasks are *PushCube*, *PickCube*. We report the mean of the success rate (%), and the best results are highlighted in **bold**. (w/ freeze) is trained with the frozen encoder, while (w/o freeze) is fine-tuning the encoder during training.

| Task \ | *PushCube* | | | *PickCube* | | |
|---|---|---|---|---|---|---|
| Algorithm | Train. | Gen. | All | Train. | Gen. | All |
| ***Stem-OB*** | **100.0** | **99.1** | **99.1** | **50.0** | **23.9** | **25.1** |
| R3M(w/ freeze) | 14.0 | 19.8 | 19.5 | 2.0 | 2.1 | 2.1 |
| R3M(w/o freeze) | 24.0 | 20.0 | 20.2 | 2.0 | 1.9 | 1.9 |
| MVP(w/ freeze) | 18.0 | 17.4 | 17.4 | 2.0 | 2.0 | 2.0 |
| MVP(w/o freeze) | 22.0 | 19.8 | 19.9 | 2.0 | 2.1 | 2.1 |

Table 16: **Success rate of SD1.4 in SAPIEN.** The tasks are *PushCube*, *PickCube*. We report the mean of the success rate (%), and the best results are highlighted in **bold**. We perform additional benchmarking on another version of stable-diffusion: SD1.4. For SAPIEN environment, the DDPM inverison steps are set to 15/50.

| Task \ | *PushCube* | | | *PickCube* | | |
|---|---|---|---|---|---|---|
| Algorithm | Train. | Gen. | All | Train. | Gen. | All |
| ***Stem-OB* (sd2.1)** | **100.0** | 99.1 | 99.1 | **50.0** | **23.9** | **25.1** |
| *Stem-OB* (sd1.4) | **100.0** | **99.7** | **99.7** | 36.0 | 18.3 | 19.1 |

While the focus of Stem-Ob's experiments is mainly on intra-class generalization, we have also conducted experiments on the ***PushCube*** task. In this task, the original cube object is replaced with 6 different and random selected object from YCB object sets (Calli et al., 2015), such as a golf ball, as is shown in the Fig. 20. We use DDPM as the inversion model in this experiment. Tab. 14 shows that more inversion steps than the previous settings do help in improving the cross-category generalization ability. While most sapien tasks exhibit an optimal inversion step of 15 steps, 20 and 25 inversion steps perform better on these cross-category settings. The success rate increases with the number of inversion steps, and Stem-OB outperforms all baselines in most settings. Due to time constraints, we have only tested cross-category generalization on the PushCube tasks so far. However, we believe this premitive experiment shows promising potential of generalization on multiple semantic hierarchies by using different inversion steps.

## I COMPARISON WITH SELF-SUPERVISED REPRESENTATION LEARNING

Self-supervised representation learning (SSL) also aims to map raw observations into a unified representation space. We benchmark our method against **R3M** Nair et al. (2022) which pre-trains a visual representation using a human video dataset and incorporates a self-supervised loss term. This representation can then be used as a perception module for downstream policy learning. We simply replace our visual encoder (ResNet18) with the pretrained R3M ResNet18 model, and then experiment with two approaches: freezing the pretrained model and fine-tuning the encoder during training. We test **R3M** on ***PushCube*** and ***PickCube***. Moreover, we try **MVP** Xiao et al. (2022), another SSL method, which is trained by masked modeling of natural images. The results are shown in the Tab. 15.

The experiment results show that the R3M pretrained encoder doesn't work well on the SAPIEN environment tasks. Since loading R3M is simple and we use the official parameters, we think this may be due to the heavy supervision imposed by R3M not suitable for diffusion policies.

## J GENERATIVE MODELS

We conduct the experiments on other generative models, other than Stable Diffusion 2.1, to find out whether the performance of *Stem-OB* is model-specific. We perform additional benchmarking on another version of stable-diffusion: SD1.4 in both MimicGen and SAPIEN environments. For MimicGen tasks, we use 1 / 8 as the inversion steps in DDPM inversion, and 15/50 for SAPIEN

Table 17: **Success rate of SD1.4 in MimicGen.** The tasks are *MugCleanup*, *Threading*, *ThreePieceAssembly*, and *Coffee*. We report the mean and standard deviation of the success rate (%), and the best results are highlighted in **bold**. We perform additional benchmarking on another version of stable-diffusion: SD1.4. For MimicGen environment, the DDPM inverison steps are set to 1/8.

| Task \ | *MugCleanup* | | | *Threading* | | | *ThreePieceAssembly* | | | *Coffee* | | |
|---|---|---|---|---|---|---|---|---|---|---|---|---|
| Algorithm | Train. | Gen. | All | Train. | Gen. | All | Train. | Gen. | All | Train. | Gen. | All |
| *Stem-OB* (sd2.1) | $\mathbf{38.7_{\pm 2.5}}$ | $\mathbf{17.1_{\pm 4.4}}$ | $\mathbf{19.5_{\pm 4.2}}$ | $31.3_{\pm 2.5}$ | $\mathbf{14.5_{\pm 3.2}}$ | $\mathbf{16.4_{\pm 3.1}}$ | $19.7_{\pm 4.8}$ | $12.3_{\pm 2.8}$ | $13.1_{\pm 3.1}$ | $\mathbf{68.0_{\pm 5.7}}$ | $\mathbf{48.0_{\pm 3.9}}$ | $\mathbf{50.5_{\pm 4.1}}$ |
| *Stem-OB* (sd1.4) | $35.3_{\pm 1.7}$ | $15.9_{\pm 4.1}$ | $18.0_{\pm 3.9}$ | $\mathbf{32.0_{\pm 0.8}}$ | $13.4_{\pm 2.6}$ | $15.5_{\pm 2.5}$ | $\mathbf{20.7_{\pm 3.9}}$ | $\mathbf{14.0_{\pm 2.2}}$ | $\mathbf{14.7_{\pm 2.4}}$ | $60.0_{\pm 6.5}$ | $37.3_{\pm 3.4}$ | $40.2_{\pm 3.9}$ |

tasks. The results of the two environments are listed in the Tab. 16 and Tab. 17. The success rates of these two versions of stable-diffusion are similar. This supports our methods' feasibility regardless of the underlying pretrained diffusion model.

