# OpenReview forum: "Stem-OB: Generalizable Visual Imitation Learning with Stem-Like Convergent Observation through Diffusion Inversion"
_ICLR.cc/2025/Conference — ICLR 2025 Spotlight_

### Official Review · Reviewer_pwGy · 2024-10-31

**Soundness:** 3
**Presentation:** 3
**Contribution:** 3
**Rating:** 8
**Confidence:** 4

**Summary:**

Visual imitation learning is a method for robots to learn tasks through visual human demonstrations. While multiple methods exist for preprocessing visual data that increase the generalizability of VIL, these methods are only partially efficient. To address the generalization issue, this paper proposes to use the inversion process of a pretrained diffusion model as a preprocessing step to increase the generalizability of visual imitation learning methods. The proposed method relies on the intuitive insight that images with only small perturbations between them are closer in latent space (and therefore become indistinguishable under diffusion faster), than data that exhibits large differences. The method is tested on real and synthetic data and shows that the proposed preprocessing significantly increases the models generalization capabilities.

**Strengths:**

* The paper relies on a central observation from a previous paper (Yue et al. 2024), which found that during the inversion process of a diffusion model, fine-grained (or high-frequency) details of an image are destroyed first before the low-level semantic concepts. This paper builds on this basic concept and proposes a new preprocessing step that uses this theoretical property of diffusion models as a preprocessing step, which, in the process, improves the generalization of the IL algorithm.
* The technical principle seems well established at this point, but this paper takes these theoretical findings and applies them to a new problem. While the whole approach is very simple from a technical standpoint, the paper re-introduces the necessary background in section 2 before developing the reader's intuition as to why this preprocessing might work in section 4.2 through theoretical explanation and small experiments. I believe that this paper warrants publication because it shows that diffusion model inversion can be useful for additional tasks.
* The presentation of the paper is clear and easy to follow overall.
* The results nicely highlight the drawbacks of previous methods and that the proposed method offers much better generalization capabilities.
* I appreciated the detailed appendix, with theoretical derivations, additional details for reproducibility, and additional experiments.

**Weaknesses:**

I believe that the paper's technical novelty is small, as the main concept has already been introduced in other papers. That being said, I also believe that the paper is worth publishing, as it adds to the growing body of literature showing the usefulness of diffusion model inversion for various tasks, and the empirical validation is well executed.

I have additional suggestions to improve the paper's readability and make it easier for the reader to understand.
1. Section 2.2 can be cut (title), and the paragraph can be integrated into Section 3.1.
2. As a reader, I would really like a short problem formulation that properly introduces the problem the paper addresses. This should properly define the inputs and outputs. I would also suggest rearranging the sections/subsections of the paper to something like this:
* Introduction
* Related Work
* Problem Definition
* Preliminaries
* Method
This would greatly improve the flow of the paper.
3. While the paper is generally easy to follow, some sentences are should be simplified or run through a writing program. I have added some examples below:
* Line 85: "To be specific, our method is as simple as inverting the image for reasonable steps before" What is reasonable? Maybe rephrase?
* Line 226: "Intuitively, given a source image and its variation, distinguishing between them becomes increasingly difficult" This sentence does not make a lot of sense (although I can guess what is meant).
There are many more of these convoluted sentences. Cleaning up the writing a bit would go a long way to improve the paper.

There

**Questions:**

See above. I currently give a weak accept to the paper but am inclined to vote for acceptance should my points above be taken into consideration during the rebuttal.

---

> ### Author Response · Authors · 2024-11-21
> **Response to reviewer pwGy**
>
> We thank you a lot for the detailed recognition of our work in the strength section and the thoughtful suggestions on the paper to make it better deliver our main idea!
>
> As the suggestions mainly focus on the writing and structure of the paper, we've made corresponding changes in the revised version, the change of the paper can be summarized as follows:
>
> 1. We rephrased and put the content of Section 2.2 to Section 3.1 as suggested, and canceled the sole subsection title of Section 2.1. This makes the preliminary part more focused on introducing the important idea of diffusion inversion.
> 2. We appreciate the suggestion and rearranged the sections accordingly, and we added a problem definition section which makes it clear that we are dealing with the visual generalization of imitation learning methods through operating on the observation before the training pipeline.
> 3. We modify line 85 to make it clear that the appropriate inversion step is selected via empirical study, balancing between a strong perturbation and better preservation of semantic information.
> 4. For line 226, we make it clear that the variation here refers to a visually altered scene of the source image that can include low-level details or high-level structures.
> 5. We also make other modifications in sections 5.1 and 5.2 along with the introduction to further clean up the writing.
>
> Thank you again for the valuable suggestions, and any further thoughts or questions are more than welcome!

---

> > ### Comment · Reviewer_pwGy · 2024-11-26
> >
> > After reading the reviews and author rebuttals, I think the paper should be accepted for publication at ICLR and raise my score to  8.

---

> ### Author Response · Authors · 2024-11-26
>
> We sincerely thank the reviewer for the thoughtful feedback and for recognizing our contributions by revising the score.

---

### Official Review · Reviewer_C5Ru · 2024-11-01

**Soundness:** 3
**Presentation:** 4
**Contribution:** 3
**Rating:** 8
**Confidence:** 3

**Summary:**

This paper proposes Stem-OB, a method that uses diffusion inversion to transform observations into a shared, robust representation. This approach enhances robustness to various appearance changes, such as shifts in visual patterns and lighting conditions. During inference, Stem-OB approximates the partial inversion process to maintain efficient inference speed.

**Strengths:**

1. The paper is well-written, with clear motivation and solid experimental validation, including both real-world and simulated experiments.

2. The central idea resonates well: robotic observations often include excessive low-level details, while effective scene understanding requires capturing high-level structural information. This paper draws on an intriguing insight from Yue et al. (2024): "Rather than uniformly removing information from different semantic hierarchies, the process brings structurally similar images closer in the early stages of inversion." Building on this, Stem-OB leverages this observation to project data into a space that prioritizes inherent structural features.

**Weaknesses:**

Stem-OB enhances the generalization capabilities of robotic systems by using diffusion inversion to map observations into a shared representation. However, another promising line of research—self-supervised representation learning—also aims to unify raw observations into a common representation space, reducing low-level visual discrepancies. I suggest that the authors consider benchmarking Stem-OB against these approaches, particularly methods that leverage self-supervised learning (SSL) for robotic representations, to offer a more comprehensive comparison.

**Questions:**

The current method conducts experiments solely on Stable Diffusion. It would be valuable to understand if the conclusions drawn from Stem-OB are applicable to other generative models, such as Flux or SD 3 (which uses flow instead of diffusion). Demonstrating that Stem-OB generalizes across different models would strengthen the claim of robustness and broaden the impact of this approach beyond a single model framework.

---

> ### Author Response · Authors · 2024-11-21
> **Response to reviewer C5Ru**
>
> We thank the reviewer for acknowledge our method's clear motivation and "solid experimental validation", and will address the concerns as follows:
>
> **Q1: Benchmarking with SSL methods.**
>
> **A1:**
> Thank you for pointing out this baseline for us! Self-supervised representation(SSL) learning also aims to unify raw observations into a common representation space.
>
> We benchmark our method against R3M [1]. It pre-trains a visual representation using the Ego4D human video dataset including a self-supervised term, which can be can be used as a perception module for downstream policy learning. We then change our visual encoder from resnet18 to the pretrained R3M resnet18 model. We have tried two ways, one is freezing the pretrained model, and the second is finetuning the encoder during training. Due to the time limit, we test R3M on ***PushCube*** and ***PickCube***. The results are shown in the table below.
>
>
> | Task | PushCube | PickCube |
> | --- | --- | --- |
> | Setting | Train. / Gen. / All | Train. / Gen. / All |
> | Stem-OB | **100.0** / **99.1±1.8** / **99.1±1.8** | **50.0** / **23.9±20.6** / **25.1±20.9** |
> | R3M(w/ freeze) | 14.0 / 19.8±6.1 / 19.5±6.1 | 2.0 / 2.1±0.8 / 2.1±0.8 |
> | R3M(w/o freeze) | 24.0 / 20.0±5.7 / 20.2±5.6 | 2.0 / 1.9±0.8 / 1.9±0.8 |
>
>
> The experiment results show that the R3M pretrained encoder doesn't work well on the SAPIEN environment tasks. Since loading R3M is simple and we use the official parameters, we think this may be due to the heavy supervision imposed by R3M not suitable for diffusion policies.
> We plan to benchmark other SSL methods like MVP[2] and will report the results once the results are available.
>
>
> [1] Nair, S., Rajeswaran, A., Kumar, V., Finn, C., & Gupta, A. (2022). R3m: A universal visual representation for robot manipulation. arXiv preprint arXiv:2203.12601.
>
> [2] Xiao, T., Radosavovic, I., Darrell, T., & Malik, J. (2022). Masked visual pre-training for motor control. arXiv preprint arXiv:2203.06173.
>
> **Q2: Other Generative Models**
>
> **A2:** Thanks for the suggestion! We perform additional benchmarking on another version of stable-diffusion, SD1.4, as we conducted some early experiments on this model other than the SD2.1 in the MimicGen environment. We then conduct additional experiments in the SAPIEN environment for further validation. For MimicGen tasks, we use 1/8 as the inversion steps in DDPM inversion, and 15/50 for SAPIEN tasks. The results of the two environments are listed in the table below.
>
>
> | Task | PushCube | PickCube |
> | --- | --- | --- |
> | Setting | Train. / Gen. / All | Train. / Gen. / All |
> | Stem-OB(sd2.1) | **100.0** / 99.1±1.8 / 99.1±1.8 | **50.0** / **23.9±20.6** / **25.1±20.9** |
> | Stem-OB(sd1.4) | **100.0** / **99.7±0.7** / **99.7±0.7** | 36.0 / 18.3±14.6 / 19.1±14.7 |
>
> | Task | MugCleanup | Threading | ThreePieceAssembly | Coffee |
> | --- | --- | --- | --- | --- |
> | Setting | Train. / Gen. / All | Train. / Gen. / All | Train. / Gen. / All | Train. / Gen. / All |
> | Stem-OB(sd2.1) | **38.7±2.5** / **17.1±11.1** / **19.5±12.5** | 31.3±2.5 / **14.5±12.3** / **16.4±12.8** | 19.7±4.8 / 12.3±6.7 / 13.1±6.9 | **68.0±5.7** / **48.0±19.7** / **50.5±19.6** |
> | Stem-OB(sd1.4) | 35.3±1.7 / 15.9±10.0 / 18.0±11.2 | **32.0±0.8** / 13.4±10.7 / 15.5±11.6 | **20.7±3.9** / **14.0±6.2** / **14.7±6.4** | 60.0±6.5 / 37.3±16.0 / 40.2±16.9 |
>
>
> As we can see in the table, the success rates of these two versions of stable-diffusion are similar. This supports our methods' feasibility regardless of the underlying pretrained diffusion model.

---

> ### Comment · Reviewer_C5Ru · 2024-11-26
>
> The authors' response has effectively addressed my concerns, so I am increasing my rating to 8.

---

> > ### Author Response · Authors · 2024-11-26
> >
> > We sincerely thank the reviewer for the thoughtful feedback and for recognizing our contributions by revising the score.

---

### Official Review · Reviewer_UAwF · 2024-11-01

**Soundness:** 3
**Presentation:** 3
**Contribution:** 4
**Rating:** 8
**Confidence:** 3

**Summary:**

This paper proposes to use diffusion inversion as input preprocessing for visual imitation learning methods in order to improve their generalization to visual changes between demonstration and test setup. The proposed approach is evaluated in real-world robotic experiments as well as on 2 simulated datasets.

**Strengths:**

- The idea is, as the authors write, 'simple yet effective' and the authors manage to neatly package a quite theoretical idea and a very practically result measured in real-world robotic experiments all in 1 paper
- While I lack the background to assess the theoretical grounding, the idea itself is well explained
- The authors combine experiments on simulated datasets with actual real-world tests. Real-world tests are incredibly important for visual methods, yet in case of robotic applications very hard to make reproducible. The combination of both is a commendable experimental design. [**update after discussion phase** the extend and rigour of experiments has even further improved during the discussion phase]
- Similarly, the authors conduct real-world user studies to investigate their hypothesis that diffusion inversion progresses along semantic hierarchies.

**Weaknesses:**

- [**update after discussion: this point was fixed**] I find the derivation in Section 4.1 very hard to follow. Only the first sentence of Section 4.2 makes it clear that the whole derivation is based on the assumption that semantically similar images are closer in latent space. That is a HUGE assumption and the following experiments on intra- and inter-class confusion do not show this over multiple levels of semantic hierarchies, but only for the semantic leafs. The introduction (lines 60-65) however argues that much more than leaf classes, abstract semantic hierarchies are important to generalize over perceptual differences.
In the empirical study on diffusion inversion, one might argue that out of the 5 empirical examples, "bowl" and "cup" are most similar. However, Table 1 shows that the latent representation of 'cup' samples is actually on average closer to any of {'drawer', 'duck', 'faucet'} than to 'bowl'. To me this makes the assumption 'semantically similar images are closer in latent space' quite unbelievable. For sure, this assumption and therefore the formulation of semantic overlap as overlap of latent gaussians is anything but 'intuitive' (line 270) in presence of this data.
- [**update after discussion: this point was fixed**] In relation to the above, I miss a clear definition of what kind of variation should be compensated through diffusion inversion. The abstract and introduction repeatedly claims that diffusion inversion can extract the 'high-level structure' of the scene, suggesting generalization over different object instances, shapes, appearances, relative placements, and robot arms. Section 4.1 considers 'variation', 'fine-grained attribute change', 'coarse-grained modifications', and 'semantic overlap' without defining any of these. The investigation in Section 4.2 is focused on variations within a semantic object class, i.e. a demonstration with 1 cup should be repeated with a different cup. The experiments then consider lighting change and a limited set of object appearance change of sometimes multiple object instances, while locations are fully fixed. The problem is that all of this currently does not fully fit together and it would be good if the authors can define more accurately what kind of variations they expect diffusion inversion to abstract / generalize over, and then design experiments accordingly to show improvement with exactly these variations.
- There are a couple of odd aspects about the simulation experiments that raise questions about the soundness of the results:
  - For the benchmarks from ManiSkill and MimicGen, why were not all tasks evaluated? For ManiSkill, plug-charger seems to be specifically excluded and for MimicGen 4 out of 12 tasks were picked without any explanation.
[ **update after discussion: now there are 8 out of 12 tasks from MimicGen. It is still a subset, but this indeed makes the results more thorough**]
  - While I did not quickly find comparable numbers for ManiSkill, the MimicGen paper reports much higher success rates for the investigated tasks. E.g. for Threading, MimicGen reports around 19% sucess from just 10 videos and 98% success from 1000 videos. For the 500 videos used in the experiments here, the success is below 19% for 3 out of 4 variants, including the proposed method. Why are the achieved success rates so low? And can the proposed method actually improve anything in a more state-of-the-art setting? [**update after discussion: This concern partially remains. While the authors show that their success rates for 500 trials are mostly between the baseline MimicGen results of 10 and 1000 trials, there is no case where the authors can demonstrate to improve over these prior results. If one expects the relation between number of trials and success rate to follow a saturation trend rather than a linear trend, their results are still a bit below those of prior works.**]
  - Why is the RO baseline excluded from the simulation experiments? [**update from discussion: this was resolved**]
- In all experiments (real and simulated), most of the differences between methods are within the standard deviation, so it is very hard to say if any conclusion can be drawn. Why is the standard deviation so high? [**update from discussion: this has been thoroughly addressed by the authors. The updated results show, similarly to the question about success rates above, that Stem-OB does not push the state-of-the-art further than where it is. But it does more importantly also not appear to be much worse, while the method in some sense is simpler.**]

**Questions:**

The most significant questions are already listed under weaknesses. In addition, I would be interested to get a clarification on the following points:

- line 272: How many images?
- line 428: Can the authors explain how they get to this conclusion? The training setting performance of RO in particular looks to me like it was not trained correctly.
- line 430: Given the result in D2B, I don't think the conclusion can be that there is always a high success rate?

---

> ### Author Response · Authors · 2024-11-21
> **Response to reviewer UAwF (1/4)**
>
> We thank the detailed and in-depth review provided by reviewer **UAwF**, we view much of the raised concerns as legitimate and conduct additional experiments to verify them, while there are cases where our results are supposedly misunderstood due to unclear statements.
>
> **Q1: Feasibility of theory assumption**
>
>
> **A1:** We appreciate the reviewer's detailed feedback and the opportunity to clarify our theoretical assumptions. The assumption that semantically similar images are closer in the latent space of pretrained diffusion models is fundamental to our approach and is supported by both prior research and empirical results.
>
> Firstly, as outlined in [1], the concept of attribute loss is inherently linked to the indistinguishability of different images during the inversion process. Attribute loss quantifies the likelihood that a diffusion model fails to distinguish noisy samples drawn from two overlapping distributions, $q(x_t|x_0)$ and $q(y_t|y_0)$. This involves:
>
> 1. **Performing diffusion inversion** on $x_0$ and $y_0$ for $t$ steps to obtain $x_t$ and $y_t$.
> 2. **Reconstructing** $x'_0$ and $y'_0$ from $x_t$ and $y_t$.
> 3. **Assessing reconstruction similarity**, where semantically similar images are more likely to produce similar reconstructions, resulting in a higher attribute loss.
>
> We adopt this refined definition in our paper (Equations (6) and (7)). By focusing on semantic leaf nodes in Section 4.2, we precisely control and measure the initial similarity between $x_0$ and $y_0$. This aligns with the formulation of attribute loss, which relies on known initial similarities and this does not mean that the distance between latent features in other semantic hierarchies is needed to fulfill the same assumption.
>
> Regarding the observations in Table 1, we acknowledge that the latent representations of 'cup' samples appear closer to some other classes than to 'bowl'. This can be attributed to the variability in visual features captured by the latent space, influenced by factors such as viewpoint, texture, and shape. While 'cup' and 'bowl' can be viewed as semantically similar, the latent space may not always reflect this due to these factors. Our main objective is to demonstrate that intra-class variations (within the same class) result in closer latent representations than inter-class variations.
>
> Importantly, the assumption that semantically similar images are closer in the diffusion model's latent space is supported by recent studies in semantic correspondence. Tang et al. [2] and Zhang et al.[3] demonstrated that pretrained diffusion models' latent spaces can effectively map semantically similar pixels across images using simple nearest-neighbor searches. This suggests that the latent space does capture semantic similarities to a significant extent.
>
> Our user study further validates this assumption by showing that semantically similar images become indistinguishable sooner during the diffusion inversion process. This empirical evidence supports the behavior predicted by attribute loss and underscores the effectiveness of our approach.
>
> We will clarify these points in the revised manuscript and provide additional evidence and references to strengthen our theoretical foundation.
>
> [1] Yue, Z., Wang, J., Sun, Q., Ji, L., Chang, E. I., & Zhang, H. (2024). Exploring diffusion time-steps for unsupervised representation learning. *arXiv preprint arXiv:2401.11430*.
>
> [2] Tang, L., Jia, M., Wang, Q., Phoo, C. P., & Hariharan, B. (2023). Emergent correspondence from image diffusion. *Advances in Neural Information Processing Systems*, 36, 1363-1389.
>
> [3] Zhang, J., Herrmann, C., Hur, J., Polania Cabrera, L., Jampani, V., Sun, D., & Yang, M. H. (2024). A tale of two features: Stable diffusion complements dino for zero-shot semantic correspondence. Advances in Neural Information Processing Systems, 36.

---

> > ### Author Response · Authors · 2024-11-21
> > **Response to reviewer UAwF (2/4)**
> >
> > **Q2: Scope of generalization**
> >
> > **A2:** Thank you for raising this important point regarding the scope of generalization. We would like to clarify that Stem-OB primarily focuses on **inter-category** visual disturbances and excels at handling high-frequency appearance changes, such as textures, patterns, and lighting conditions. These types of disturbances are effectively mitigated because they are attenuated earlier in the diffusion inversion process.
> >
> > The terms "fine-grained" and "coarse-grained" are inspired by the theoretical basis proposed in DiTi [1] (see Section 4.1 of the original paper), which examines pixel-level changes during diffusion inversion. For example, in human facial images, fine-grained details such as expressions are eliminated in early inversion steps, while coarse-grained attributes such as gender persist until later steps. This observation underpins our tree-like structural representation of inversion-induced observations.
> >
> > The intra-category variance (e.g., differences between cups) is essential for defining task-relevant information. In contrast, generalization across categories (e.g., from cups to bowls) is less meaningful because the task objectives differ fundamentally between categories.
> >
> > That being said, we argue that the distinction between "inside" and "outside" generalization scope is more applicable to augmentation-based methods than to ours. In augmentation methods, variations are explicitly collected or generated and added to the training data. This approach provides generalization capability for the specific types of variations included but lacks robustness to unseen variations. In contrast, our method removes unnecessary information from observations and adapts to different variations without pre-defining the generalization scope, making it more versatile and aligned with real-world needs.
> >
> > To validate this claim, we introduced a new type of visual variance in the SAPIEN environment by adding normal maps to the tabletop and combining this perturbation with other variations, such as lighting conditions and table textures. The results demonstrate that Stem-OB consistently outperforms the baselines, even when encountering previously unseen perturbations. A visualization of this experiment is included in the appendix.
> >
> > | Task | PushCube | PickCube |
> > | --- | --- | --- |
> > | Setting | normal only / normal+texture / normal+light  | normal only / normal+texture / normal+light  |
> > | Stem-OB | ***98.0±2.3*** / **99.3±0.9** / ***98.7±0.9*** | **30.0±21.5** / **2.3±0.8** / **17.7±21.5** |
> > | Org | 74.7±13.4 / 39.0±1.5 / 58.5±27.1 | 2.0±0.0/  ***2.0±0.0*** / 2.0±0.0 |
> > | SRM | 86.0±8.1 / 8.3±2.7 / 75.8±11.7 | 2.3±0.8 / **2.3±0.8** / 2.2±0.5 |
> > | Mix | **99.0±1.0** / ***97.3±1.9*** / **99.0±1.0** | ***22.3±17.5*** / ***2.0±0.0*** / ***8.5±5.9*** |
> >
> > We'd like to note that there seems to be a misunderstanding in describing our experiment locations as "fully fixed". In the simulation tasks, all test-time object positions are randomized within a fixed area defined by MimicGen or SAPIEN. For real-world tasks, each generalization setting is evaluated in nine distinct, randomly selected positions covering the training distribution. For fairness, these testing positions are fixed across different generalization settings and baselines.

---

> > > ### Author Response · Authors · 2024-11-21
> > > **Response to reviewer UAwF (3/4)**
> > >
> > > **Q3: Simulation experiments and standard variation**
> > >
> > > **A3:** We believe there is a misunderstanding regarding our results, which we are happy to clarify. The high standard deviation reported in some of our experiments, as long as the fact that the performance doesn't match reported data in MimicGen paper results from the fact that we average success rates across diverse environment variation settings. Specifically, this includes both the training environment, where the policy performs well and matches the original data, and several challenging test variants, where performance can vary significantly. Aggregating results across these diverse settings naturally results in larger standard deviations.
> > >
> > > We have now separated the results for training and test variants to provide a clearer picture of the policy's performance. For SAPIEN tasks, we test 50 episodes for each task, while for MimicGen tasks, we test 300 episodes.
> > >
> > > | Task | PushCube | PegInsertionSide | PickCube | StackCube |
> > > | --- | --- | --- | --- | --- |
> > > | Setting | Train. / Gen. / All | Train. / Gen. / All | Train. / Gen. / All | Train. / Gen. / All |
> > > | Stem-OB | **100.0** / **99.1±1.8** / **99.1±1.8** | **2.0** / ***0.3±0.7*** / ***0.4±0.8*** | 50.0 / **23.9±20.6** / **25.1±20.9** | **90.0** / **24.1±28.7** / **27.2±31.3** |
> > > | Org | **100.0** / 59.9±32.7 / 61.8±33.1 | 0.0 / 0.0±0.0 / 0.0±0.0 | **90.0** / 4.6±1.1 / 8.6±18.2 | **90.0** / 0.0±0.0 / 4.3±19.2 |
> > > | SRM | **100.0** / 60.6±30.0 / 62.5±30.5 | 0.0 / 0.0±0.0 / 0.0±0.0 | ***80.0*** / 4.8±1.6 / 8.4±16.1 | 16.0 / 0.0±0.0 / 0.7±3.4 |
> > > | Mix | **100.0** / ***97.1±3.4*** / ***97.2±3.4*** | **2.0** / **0.5±1.2** / **0.6±0.1** | 75.0 / ***9.8±8.8*** / ***12.9±16.3*** | ***84.0*** / ***4.4±6.3*** / ***8.2±18.0*** |
> > >
> > > | Task | MugCleanup | Threading | ThreePieceAssembly | Coffee |
> > > | --- | --- | --- | --- | --- |
> > > | Setting | Train. / Gen. / All | Train. / Gen. / All | Train. / Gen. / All | Train. / Gen. / All |
> > > | Stem-OB | ***38.7±2.5*** / ***17.1±11.1*** / ***19.5±12.5*** | 31.3±2.5 / 14.5±12.3 / 16.4±12.8 | 19.7±4.8 / 12.3±6.7 / 13.1±6.9 | **68.0±5.7** / **48.0±19.7** / **50.5±19.6** |
> > > | Org | 33.0±2.8 / 16.4±12.0 / 16.4±12.0 | **41.3±3.3** / 15.2±10.4 / 18.1±12.8 | **26.3±4.5** / ***13.4±7.7*** / **14.8±8.4** | ***61.0±6.5*** / ***40.7±16.8*** / ***43.3±17.3*** |
> > > | SRM | 21.0±2.9 / 15.6±7.1 / 15.4±0.1 | 24.3±5.7 / **26.0±12.2** / **25.8±11.7** | ***24.7±2.9*** / 12.3±6.8 / 13.6±7.6 | 52.3±3.7 / 37.6±11.7 / 39.4±12.0 |
> > > | Mix | **39.7±6.9** / **21.9±12.3** / **22.3±11.8** | ***32.0±1.6*** / ***17.3±7.5*** / ***18.9±8.5*** | 17.7±1.7 / **13.8±6.8** / ***14.2±6.6*** | 55.3±5.3 / 39.8±8.8 / 41.8±9.9 |
> > >
> > > To further strengthen our evaluation, we added experiments for additional MimicGen tasks. While our initial selection of 4 tasks out of 12 was somewhat arbitrary, this practice is common in visual imitation learning research. For instance, recent works on MimicGen have used 4 tasks in [1], 4 in [2], 6 in [3], 8 in [4], and 9 in [5]. Due to time constraints, we report results for an additional 4 tasks here, leaving the rest for future studies. The additional tasks include ***Square_D0***, ***StackThree_D0***, ***Stack_D0***, and ***CoffeePreparation_D0***. We report success rates separately for training and generalization settings. The results demonstrate that **Stem-OB** achieves performance that is either superior to or on par with all baselines.
> > >
> > > | Task | Square | StackThree | Stack | CoffeePreparation |
> > > | --- | --- | --- | --- | --- |
> > > | Setting | Train. / Gen. / All | Train. / Gen. / All | Train. / Gen. / All | Train. / Gen. / All |
> > > | Stem-OB | 50.7±1.7 / ***28.3±12.8*** / ***31.4±14.0*** | ***41.3±0.9*** / **27.6±12.3** / **29.3±12.4** | ***85.7±0.9*** / ***70.7±29.0*** / ***72.6±27.5*** | 14.7±2.1 / **13.2±6.4** / **13.4±6.0** |
> > > | Org | ***53.3±5.2*** / 22.0±12.5 / 25.9±15.7 | **44.0±5.1** / ***25.7±15.1*** / ***28.0±15.5*** | **87.3±2.1** / **74.5±29.3** / **76.1±27.7** | **18.7±5.0** / 10.1±5.9 / 11.1±6.5 |
> > > | SRM | 51.7±0.5 / 22.4±13.0 / 26.1±15.6 | 28.0±0.0 / 20.4±9.1 / 21.3±8.9 | 75.3±3.4 / 58.7±24.8 / 60.8±23.8 | 15.0±2.2 / 9.5±5.2 / 10.2±5.2 |
> > > | Mix | **53.7±5.6** / **37.5±15.4** / **39.5±15.5** | 35.7±2.5 / 18.3±9.2 / 20.5±10.4 | 79.7±3.1 / 66.8±27.6 / 68.4±26.2 | ***17.7±2.5*** / ***11.9±6.0*** / ***12.6±6.0*** |

---

> > > > ### Author Response · Authors · 2024-11-21
> > > > **Response to reviewer UAwF (4/4)**
> > > >
> > > > Regarding the SAPIEN tasks, since the dataset is originally state-based, we replayed the trajectories and collected a visual imitation learning dataset. We excluded the ***PlugCharger*** task because it is challenging to find an informative camera position, and all baselines perform poorly on this task.
> > > >
> > > > Finally, regarding the exclusion of the Random Overlay (**RO**) baseline in some experiments, the original RO baseline uses the Places365 dataset, which contains real-world images inconsistent with the simulation environments, thus we didn't take it into account in simulation tasks. To adapt it for simulation, we constructed a simulation dataset by overlaying images from the MimicGen dataset. Specifically, we randomly selected one of the seven MimicGen environments and picked an arbitrary image from a random trajectory to overlay onto the original image. Using this adapted RO dataset, we evaluated **RO** on ***PushCube***, ***PickCube*** and ***StackCube***. The results show that **Stem-OB** still outperforms **RO** in most tasks.
> > > >
> > > > | Task | PushCube | PickCube | StackCube |
> > > > | --- | --- | --- | --- |
> > > > | Setting | Train. / Gen. / All | Train. / Gen. / All | Train. / Gen. / All |
> > > > | Stem-OB | **100.0** / **99.1±1.8** / **99.1±1.8** | 50.0 / **23.9±20.6** / **25.1±20.9** | **90.0** / **24.1±28.7** / **27.2±31.3** |
> > > > | RO | **100.0** / 92.1±18.0 / 92.5±17.7 | **86.0** / 22.5±20.9 / **25.5±24.5** | 60.0 / 7.2±11.5 / 9.7±15.9 |
> > > >
> > > > [1] Wang, R., Zhuang, Z., Jin, S., Ingelhag, N., Kragic, D., & Pokorny, F. T. (2024). Feature Extractor or Decision Maker: Rethinking the Role of Visual Encoders in Visuomotor Policies. arXiv preprint arXiv:2409.20248.
> > > >
> > > > [2] Wang, D., Hart, S., Surovik, D., Kelestemur, T., Huang, H., Zhao, H., ... & Platt, R. (2024). Equivariant diffusion policy. arXiv preprint arXiv:2407.01812.
> > > >
> > > > [3] Tian, S., Wulfe, B., Sargent, K., Liu, K., Zakharov, S., Guizilini, V., & Wu, J. (2024). View-invariant policy learning via zero-shot novel view synthesis. arXiv preprint arXiv:2409.03685.
> > > >
> > > > [4] Wang, Y., Zhang, Y., Huo, M., Tian, R., Zhang, X., Xie, Y., ... & Tomizuka, M. (2024). Sparse diffusion policy: A sparse, reusable, and flexible policy for robot learning. arXiv preprint arXiv:2407.01531.
> > > >
> > > > [5] Jiang, Z., Xie, Y., Lin, K., Xu, Z., Wan, W., Mandlekar, A., ... & Zhu, Y. (2024). DexMimicGen: Automated Data Generation for Bimanual Dexterous Manipulation via Imitation Learning. arXiv preprint arXiv:2410.24185.
> > > >
> > > > **Q4: How many images were used in the illustrative experiment**
> > > >
> > > > **A4:** We use 4 images for each category. These images are the same as those used in the disturbance setting of real-world tasks, an overview of these images can be found in Figure 13, we've also made it more clear in the revised paper about the amount and source of those images.
> > > >
> > > > **Q5: Real-world training setting performance**
> > > >
> > > > **A5:** We use the expression of "relatively high performance" in contrast to the generative performances to emphasize the performance drop of most baselines in the face of visual disturbances. The evaluation of the training setting is conducted in the same setting of the original dataset, and the baselines will apply different changes to the dataset in trying to improve the policy's robustness, while for Random Overlay (**RO**), it is possible that the introduced random real-world image overlay during training stage will hinder the performance of the model, making the training performance lower than the **Org** setting.
> > > >
> > > > **Q6: High success rate wording**
> > > >
> > > > **A6:** Thank you for pointing this out, while the success rate of 44.4% may not be accounted as a "high success rate", the intention here is to compare with the other baselines with a success rate of around 10%-20%, which our method clearly stands out. We've rephrased the sentence to make it clear we are comparing with the baselines.

---

> > > ### Comment · Reviewer_UAwF · 2024-11-25
> > > **does not fit method  name**
> > >
> > > To provide quick feedback to the authors: The arguments seem more fitting to the paper experiments on first glance. However, reducing the paper to compensate for intra-category variations requires renaming the method. The name currently is motivated by a stem in a hierarchical tree. Unless the authors provide arguments that the diffusion inversion captures multi-level semantic hierarchies, the name does not make sense anymore.

---

> ### Author Response · Authors · 2024-11-26
> **Clarification for renaming request**
>
> Dear reviewer UAwF:
> Thank you for your reply, we may ask for some clarifications on your comment.
>
> Generally speaking, do you think the current name is not appropriate since it implies our method has the ability to perform generalization in "multi-level semantic hierarchies", which it doesn't?
> If so, there are a few questions:
> 1. Can you give us a few examples to better understand the meaning of multi-level semantic hierarchies here? For example, at the base level, there are different types of cups and at the medium level, there are cups and bowls, etc.
> 2. Do you think the ability of inter-category generalization is the key to proving that our method can generalize across different semantic hierarchies? If so, could an additional experiment about, for example, generalizing to pick up cups from demonstrations of picking up bowls prove such ability?
> 3. What level of semantic hierarchy do you wish to see for generalization, since we have shown the ability to generalize across different lighting conditions, different object surface patterns, and different normal maps (equivalent to the shadow over the surface)
> 4. Lastly, could you please give us a hint on which part of the current title you think is the most inappropriate? Maybe a change in the paper title could better deliver our contribution.
>
> We would like to argue that the different lighting conditions and the appearance of the same category of object are the most useful variations in practice, which Stem-Ob has already demonstrated through simulation and real-world experiments. If we are generalizing across categories, the definition of the task becomes ambiguous. For example, putting a lid on a cup is not similar to putting a lid on a bowl, thus making it less useful in real tasks. While Stem-Ob could potentially perform such kind of generalization given more inversion steps, we think the upper level of the semantic tree is the most worthy to generalize since these objects share the same function and can be easily generalized by humans.
>
> Thank you for your comment and they are of great value for us to improve the paper.

---

> > ### Comment · Reviewer_UAwF · 2024-11-26
> > **Answer: Clarification for renaming request**
> >
> > Thank you authors for providing that additional input. I guess your argument in 3 that different lighting conditions etc are already part of a tree makes sense and I assume it was mostly due to my background that I connected a tree only with class hierarchies. I re-checked the manuscript and I think Figure 1 already introduces it well, so that is an error on my part and it can be indeed already considered a tree with multiple levels even for a single level of classification hierarchy.

---

> > > ### Author Response · Authors · 2024-11-27
> > >
> > > Thank you for taking the time to re-evaluate our arguments and for acknowledging the validity of our explanation. We appreciate your engagement with our work and your recognition that Figure 1 effectively introduces this concept.
> > >
> > > We hope that this clarification, along with the additional revisions we have made to address other concerns, provides confidence in the contribution and soundness of our work. If you feel these improvements sufficiently address your previous reservations, we kindly invite you to reconsider your score to 6 (borderline accept), as we believe this work aligns with the expectations of ICLR for advancing the field.

---

> > > > ### Comment · Reviewer_UAwF · 2024-11-28
> > > > **Summary of Weaknesses / Concerns after Author Response**
> > > >
> > > > I finally had time to go through the extensive answer by the authors. Their answers definitely addressed some concerns and were even able to resolve some. I particularly want to thank the authors for taking so much time to explain their settings and answer all the questions I had.
> > > > In the interest of overview, here is a summary of the concerns that were not addressed by the authors:
> > > >
> > > > - The authors clarified that the proposed method should only be applied to generalise over intra-class variations. E.g. for a task such as "poring water", exchanging a cup with a bowl will not work. However, changing lighting conditions, positions, and some other forms of appearance can work. I have to find time to re-read the revised manuscript to make sure that this cannot be misunderstood anymore, but I assume right now that it is fine and simply a bit smaller scope than I initially understood.
> > > > - The evaluation is still run over picked subsets of the overall tasks: 4 out of 5 for ManiSkill and 8 out of 12 (previously 4 out of 12) for MimicGen. The authors however add the explanation that the other tasks perform poorly for all methods. That comment should maybe be added to the manuscript as it is important information regarding the complexity of the tasks that the proposed method can handle.
> > > > - The authors explain better why the standard deviation is so high. It however does not change that most of the results are not statistically significant, where the difference between the proposed method and other methods is within one sigma.
> > > > - The concern about the overall low success rates has not been addressed: "While I did not quickly find comparable numbers for ManiSkill, the MimicGen paper reports much higher success rates for the investigated tasks. E.g. for Threading, MimicGen reports around 19% sucess from just 10 videos and 98% success from 1000 videos. For the 500 videos used in the experiments here, the success is below 19% for 3 out of 4 variants, including the proposed method. Why are the achieved success rates so low? And can the proposed method actually improve anything in a more state-of-the-art setting?"
> > > >
> > > > My current takeaway given this summary is that the paper studies an interesting idea. It is not clear from the paper whether this idea is really helpful or will have impact on the field, because the utility compared to other approaches is very much up for debate. However, the study itself seems to me well enough conducted to share the idea with the ICLR community. I will as pointed out re-check the claims and description of the generalisation and performance of the method in the revision, but if that checks out plan to raise my score to borderline accept.

---

> > > > > ### Author Response · Authors · 2024-11-30
> > > > > **Reply to Summary of Weaknesses / Concerns (1/3)**
> > > > >
> > > > > Thank you for your detailed analysis of the current weakness and concerns! We believe among the four points raised, the Q3 and Q4 are largely due to unclear statements. We make a more detailed explanation of the selection of the current tasks for Q2 and conduct additional experiments for Q1 to demonstrate some cross-category generalization ability that aligns well with our theory.
> > > > >
> > > > > **Q1. Generalization Scope**
> > > > >
> > > > > While the focus of Stem-Ob's experiments is mainly on intra-class generalization, we have conducted new experiments on SAPIEN **PushCube** tasks that perfectly match our theory. We replace the original cube object with 8 different and randomly selected objects from YCB object sets [1], such as a golf ball. The experimental results show that more inversion steps than the previous settings do help in improving the cross-category generalization ability.
> > > > >
> > > > > While most SAPIEN tasks exhibit an optimal inversion step of 15 steps, 20 and 25 inversion steps perform better in these cross-category settings. As shown in the table below, the success rate increases with the number of inversion steps, and Stem-OB outperforms all baselines in most settings.
> > > > >
> > > > > Due to time constraints, we have only tested cross-category generalization on the **PushCube** tasks. However, we believe this primitive experiment shows promising potential for generalization on multiple semantic hierarchies by using different inversion steps, and we plan to update the results in the final version of the paper.
> > > > >
> > > > > |Settings|50_15_ddpm|50_20_ddpm|50_25_ddpm|org|srm|mix|
> > > > > |---|---|---|---|---|---|---|
> > > > > |chef_can|0.30|**0.34**|**0.34**|0.30|0.26|0.28|
> > > > > |dice|**0.20**|**0.20**|**0.20**|0.16|**0.20**|0.24|
> > > > > |golf|**0.30**|0.26|**0.30**|0.24|0.24|0.26|
> > > > > |orange|0.34|0.42|**0.44**|0.32|0.34|0.42|
> > > > > |soccer|0.30|**0.40**|**0.40**|0.28|0.30|0.34|
> > > > > |fish_can|0.24|**0.34**|**0.34**|0.20|0.28|**0.34**|
> > > > > |strawberry|0.20|0.24|**0.28**|0.16|**0.28**|0.26|
> > > > > |pear|0.24|**0.34**|**0.34**|0.16|0.28|0.32|
> > > > >
> > > > > [1] Berk Calli, Aaron Walsman, Arjun Singh, Siddhartha Srinivasa, Pieter Abbeel, and Aaron M. Dollar, Benchmarking in Manipulation Research: The YCB Object and Model Set and Benchmarking Protocols, IEEE Robotics and Automation Magazine, pp. 36 – 52, Sept. 2015.
> > > > >
> > > > > **Q2. Simulation tasks**
> > > > >
> > > > > For the SAPIEN tasks, the available dataset on the internet is state-based, so we had to replay the trajectories and collect a visual imitation learning dataset. This required manually selecting the camera positions. In the ***PlugCharger*** environment, the task itself is quite challenging, and the gripper obstructs the view from most directions when inserting the plug into the socket. As a result, training a good policy using just a single camera position proved difficult. Due to these constraints, we decided to exclude the ***PlugCharger*** task from our experiments.
> > > > >
> > > > > For the MimicGen tasks, we didn't use these four tasks: ***Pick Place***, ***Nut Assembly***, ***Hammer Cleanup***, and ***Kitchen***. The first two tasks involve a Sawyer robot arm, instead of a more common Franka arm in the dataset. And we face some technical issues in successfully replaying the dataset to collect and test observations of different resolutions. As for ***Hammer Cleanup*** and ***Kitchen***, we encountered issues with unregistered environments, likely due to updates made to MimicGen after we conducted most of our experiments. We decided to shift our focus more to the SAPIEN tasks because it offers higher-fidelity visual and physical simulations that more closely resemble real-world conditions. As a result, we skipped these environments in MimicGen. We will address these issues in detail in the next version of our manuscript.
> > > > >
> > > > > Also, it’s worth noting that our initial selection of 4 tasks out of 12 was not an uncommon practice. For instance, recent robotic learning works using the MimicGen benchmark used 4 tasks in [1], 4 in [2], 6 in [3], 8 in [4], and 9 in [5], without taking all the 12 tasks in the experiments.
> > > > >
> > > > > [1] Wang, R., Zhuang, Z., Jin, S., Ingelhag, N., Kragic, D., & Pokorny, F. T. (2024). Feature Extractor or Decision Maker: Rethinking the Role of Visual Encoders in Visuomotor Policies. arXiv preprint arXiv:2409.20248.
> > > > >
> > > > > [2] Wang, D., Hart, S., Surovik, D., Kelestemur, T., Huang, H., Zhao, H., ... & Platt, R. (2024). Equivariant diffusion policy. arXiv preprint arXiv:2407.01812.
> > > > >
> > > > > [3] Tian, S., Wulfe, B., Sargent, K., Liu, K., Zakharov, S., Guizilini, V., & Wu, J. (2024). View-invariant policy learning via zero-shot novel view synthesis. arXiv preprint arXiv:2409.03685.
> > > > >
> > > > > [4] Wang, Y., Zhang, Y., Huo, M., Tian, R., Zhang, X., Xie, Y., ... & Tomizuka, M. (2024). Sparse diffusion policy: A sparse, reusable, and flexible policy for robot learning. arXiv preprint arXiv:2407.01531.
> > > > >
> > > > > [5] Jiang, Z., Xie, Y., Lin, K., Xu, Z., Wan, W., Mandlekar, A., ... & Zhu, Y. (2024). DexMimicGen: Automated Data Generation for Bimanual Dexterous Manipulation via Imitation Learning. arXiv preprint arXiv:2410.24185.

---

> > > > > > ### Author Response · Authors · 2024-11-30
> > > > > > **Reply to Summary of Weaknesses / Concerns (2/3)**
> > > > > >
> > > > > > **Q3. Standard deviation**
> > > > > >
> > > > > > Thank you for pointing out the issue with the standard deviation in our previous report. We apologize for the confusion caused by the large reported standard deviation and believe this is largely due to our inappropriate calculation method. We will now explain a more suitable way and why the previous standard deviation results are not that meaningful.
> > > > > >
> > > > > > For Mimicgen tasks, consider one single task. There are $n$ kinds of visual generalization variations $v_i, i=1,...,n$. In our previous report, we conducted 300 trials on each setting and pooled all the success rates, along with the performance on the original environment, together to calculate a single mean and variance. This approach was flawed because the success rates from different settings vary a lot, leading to a misleadingly large and uninformative standard deviation.
> > > > > >
> > > > > > We'll introduce a more meaningful analysis here, for each setting $v_i$, we divided the 300 episodes into 3 groups and calculated the success rate means $\mu_i$ and standard deviation $\sigma_i$ between each group. The overall mean for each task is the average of the individual means $\mu=\frac{1}{n} \Sigma_{i=1}^{n} \mu_i$, and the overall standard deviation is computed as the square root average across all settings $\sigma = \sqrt{\frac{1}{n}\Sigma_{i=1}^{n} \sigma_i^2}$. This method more accurately reflects the fluctuation of success rates across the different settings. The updated table of MimicGen success rates is provided below and we witness a large drop in standard deviation from ~±20 to ~±5.
> > > > > >
> > > > > >
> > > > > > | Task | MugCleanup | Threading | ThreePieceAssembly | Coffee |
> > > > > > | --- | --- | --- | --- | --- |
> > > > > > | Setting | Train. / Gen. / All | Train. / Gen. / All | Train. / Gen. / All | Train. / Gen. / All |
> > > > > > | Stem-OB | ***38.7±2.5*** / ***17.1±4.4*** / ***19.5±4.2*** | 31.3±2.5 / 14.5±3.2 / 16.4±3.1 | 19.7±4.8 / 12.3±2.8 / 13.1±3.1 | **68.0±5.7** / **48.0±3.9** / **50.5±4.1** |
> > > > > > | Org | 33.0±2.8 / 14.4±3.5 / 16.4±3.4 | **41.3±3.3** / 15.2±2.4 / 18.1±2.5 | **26.3±4.5** / ***13.4±1.8*** / **14.8±2.3** | ***61.0±6.5*** / ***40.7±4.1*** / ***43.3±4.5*** |
> > > > > > | SRM | 21.0±2.9 / 14.7±3.0 / 15.4±3.0 | 24.3±5.7 / **26.0±4.4** / **25.8±4.6** | ***24.7±2.9*** / 12.3±2.4 / 13.6±2.5 | 52.3±3.7 / 37.6±2.7 / 39.4±2.9 |
> > > > > > | Mix | **39.7±6.9** / **20.1±3.0** / **22.3±3.6** | ***32.0±1.6*** / ***17.3±2.1*** / ***18.9±2.1*** | 17.7±1.7 / **13.8±2.2** / ***14.2±2.2*** | 55.3±5.3 / 39.8±3.8 / 41.8±4.0 |
> > > > > >
> > > > > >
> > > > > > | Task | Square | StackThree | Stack | CoffeePreparation |
> > > > > > | --- | --- | --- | --- | --- |
> > > > > > | Setting | Train. / Gen. / All | Train. / Gen. / All | Train. / Gen. / All | Train. / Gen. / All |
> > > > > > | Stem-OB | 50.7±1.7 / ***28.3±3.5*** / ***31.4±3.3*** | ***41.3±0.9*** / **27.6±1.8** / **29.3±1.7** | ***85.7±0.9*** / ***70.7±2.8*** / ***72.6±2.6*** | 14.7±2.1 / **13.2±1.8** / **13.4±1.8** |
> > > > > > | Org | ***53.3±5.2*** / 22.0±3.1 / 25.9±15.7 | **44.0±5.1** / ***25.7±1.6*** / ***28.0±3.5*** | **87.3±2.1** / **74.5±2.5** / **76.1±2.5** | **18.7±5.0** / 10.1±1.6 / 11.1±2.3 |
> > > > > > | SRM | 51.7±0.5 / 22.4±3.8 / 26.1±3.6 | 28.0±0.0 / 20.4±1.8 / 21.3±1.7 | 75.3±3.4 / 58.7±4.0 / 60.8±3.8 | 15.0±2.2 / 9.5±2.9 / 10.2±2.8 |
> > > > > > | Mix | **53.7±5.6** / **37.5±4.7** / **39.5±4.8** | 35.7±2.5 / 18.3±3.2 / 20.5±3.1 | 79.7±3.1 / 66.8±3.8 / 68.4±3.7 | ***17.7±2.5*** / ***11.9±2.8*** / ***12.6±2.8*** |

---

> > > > > > > ### Author Response · Authors · 2024-11-30
> > > > > > > **Reply to Summary of Weaknesses / Concerns (3/3)**
> > > > > > >
> > > > > > > For the SAPIEN tasks, we face a similar problem in reporting the average and standard deviation values across the generalization settings. Due to time constraints, we were only able to test 50 episodes for each setting in each task. As a result, we did not have enough data to group the results and compute the mean and standard deviation for each setting. However, the reported average success rates are **correct and provide useful insights**. We can use hypothesis testing to demonstrate that Stem-OB significantly outperforms all the baselines.
> > > > > > >
> > > > > > > Suppose the success rate in the given setting for a specific task is drawn from unknown geometric distributions with unknown parameters $p$. We have observed the success rates from each group over $n$ samples. Let $p_1$ represent Stem-OB's success rate, and $p_2$ represent the success rate of a baseline. We aim to test the null hypothesis $H_0: p_1 \le p_2$, which is a one-tailed test. The standard error of the difference between two independent proportions is given by: $SE(D)=\sqrt{\frac{\hat{p_1}(1-\hat{p_1})}{n} + \frac{\hat{p_2}(1-\hat{p_2})}{n}}$, where $D=\hat{p_1}-\hat{p_2}$. So The test statistic for this hypothesis test is calculated as the difference divided by the standard error $z=\frac{D}{SE(D)}$. To perform the hypothesis test at a 95% confidence level, the critical value for a one-tailed test is 1.645. For the null hypothesis to be rejected (i.e., to conclude that $p_1 > p_2$), the test statistic must exceed this critical value.
> > > > > > >
> > > > > > > We calculate the z-value of Stem-OB for each setting against all three baselines. The **significant count** table is presented as $n/m$, where $n$ represents the number of instances where $z>1.645$, which means Stem-OB significantly outperforms the corresponding baseline, and $m$ represents the number of instances where $z<-1.645$, indicating Stem-OB significantly underperform the corresponding baseline.
> > > > > > >
> > > > > > > According to the table, Stem-OB significantly outperforms the baselines in most settings across these tasks, with very few or no instances of underperformance. Therefore, we can conclude that Stem-OB demonstrates superior generalization ability compared to the baselines, with statistical significance.
> > > > > > >
> > > > > > > We sincerely thank you for this suggestion, we will introduce the more meaningful standard deviations throughout the paper and include the hypothesis test in the updated version of our manuscript.
> > > > > > >
> > > > > > > **significant count**
> > > > > > >
> > > > > > > | Tasks           | org_z   | srm_z   | mix_z   |
> > > > > > > |:---------------|:--------|:--------|:--------|
> > > > > > > | PushCube  | 14/0  | 16/0 | 4/0  |
> > > > > > > | PegInsertionSide | 0/0  | 0/0  | 0/0  |
> > > > > > > | PickCube | 11/1  | 11/1  | 9/1  |
> > > > > > > | StackCube |  11/0 | 12/0  | 11/0 |
> > > > > > >
> > > > > > > **Q4. low success rates**
> > > > > > >
> > > > > > > Thank you for pointing out this issue, and we believe the confusion is caused by our unclear reporting. As stated in Q3, the original success rates we reported were averaged across all settings, which includes both the training environment—where the policy performs well and aligns with the original results, and several challenging test variants, where performance can vary significantly. Aggregating results across such diverse settings leads to larger standard deviations and lower average success rates.
> > > > > > >
> > > > > > > We have separated the success rates for the training setting and generalization setting, the result this is also presented in Table 9 of the revised paper. To further clarify that Stem-OB and the baselines perform well in the original setting (i.e., the training setting), we provide the success rates for all 8 MimicGen tasks under the training setting. The success rates for 10 demos and 1000 demos are taken from the MimicGen paper [1].
> > > > > > >
> > > > > > > As shown in the table below, the success rate of **Org** is generally between the 10-demo and 1000-demo results, except for Coffee, this discrepancy may be due to the fact that the 10 demos success rate for **Coffee** is exceptionally high compared to other tasks. The **CoffeePreparation** task is a long-horizon task, so **Org** with 500 demos doesn't improve so much.
> > > > > > >
> > > > > > > The results have indicated that Stem-OB and other baselines can achieve a reasonable success rate on the training setting, consistent with the results of the original MimicGen paper.
> > > > > > >
> > > > > > > | Task  | 10 demos (Source) | 500 demos (our Org) | 1000 demos (D0) |
> > > > > > > |---|-------|------------|------------|
> > > > > > > | MugCleanup | 12.7±2.5 | 33.0±2.8 | 80.0±4.9 |
> > > > > > > | Threading | 19.3±3.4 | 41.3±3.3 | 98.0±1.6 |
> > > > > > > | ThreePieceAssembly | 1.3±0.9 | 26.3±4.5 | 82.0±1.6 |
> > > > > > > | Coffee | 74.0±4.3 | 61.0±6.5 | 100.0±0.0 |
> > > > > > > | Square | 11.3±0.9 | 53.3±5.2 | 90.7±1.9 |
> > > > > > > | StackThree | 0.7±0.9 | 44.0±5.1 | 92.7±1.9 |
> > > > > > > | Stack | 26.0±1.6 | 87.3±2.1 | 100.0±0.0 |
> > > > > > > | CoffeePreparation | 12.7±3.4 | 18.7±5.0 | 97.3±0.9 |
> > > > > > >
> > > > > > > [1] Mandlekar, A., Nasiriany, S., Wen, B., Akinola, I., Narang, Y., Fan, L., ... & Fox, D. (2023). Mimicgen: A data generation system for scalable robot learning using human demonstrations. arXiv preprint arXiv:2310.17596.

---

> > > > > > > > ### Comment · Reviewer_UAwF · 2024-12-02
> > > > > > > > **Thanks to authors**
> > > > > > > >
> > > > > > > > Dear authors,
> > > > > > > >
> > > > > > > > I fear I have been quite a pain to you during this discussion (and indeed in the openreview order I seem to be reviewer 2...), but I sincerely thank you for all the answers and explanations. It helped me to understand your results much better and assess it more accurately. After checking the revised manuscript again, I have raised my score to accept.
> > > > > > > >
> > > > > > > > There are two final comments that I want to put here in case that the paper is overall accepted, to hopefully help you improve the final manuscript.
> > > > > > > >
> > > > > > > > - When I reread the text, I noticed that what confused me a lot was the term "high-level semantics" that is used throughout the text. I think what the authors mean is "a high level abstraction of individual appearances, i.e. semantics". How I instead understood this phrase is "a high level abstraction of individual semantic categories, i.e. the overall cluster of vessels that includes cups and bowls". Maybe finding a different formulation will avoid this confusion for other readers.
> > > > > > > > - For the statistical test of the SAPIEN results, it would help to somewhere report the number of settings. If I understood correctly, it is 21 as reported in appendix C2. If that number is correct, that would mean that the proposed method does not outperform baselines in the majority of settings, so the number of settings is relevant to report.

---

> > > > > > > > > ### Author Response · Authors · 2024-12-03
> > > > > > > > > **Acknowledgment of Reviewer’s Efforts**
> > > > > > > > >
> > > > > > > > > Dear reviewer UAwF:
> > > > > > > > >
> > > > > > > > > We sincerely thank you for your insightful opinion, meticulous feedback, and careful attention to details throughout the review process. Your comments have helped us address unclear statements and greatly improve the quality and clarity of our paper. We deeply appreciate the time and care you dedicated to the reviewing process, and feel incredibly fortunate to have had a "reviewer 2" like you.
> > > > > > > > >
> > > > > > > > > Regarding the final comments, we admit that finding a precise expression of the "semantics level" is important and challenging. What we want to express is indeed focusing on the abstraction of individual appearances, we'll try to polish relevant parts in the final version. About the second part, what we want to show here is a clear advantage of significantly outperforming count $n$ over significantly underperforming number $m$, and this is true for all the tasks against the best baseline "mix" except the "PegInsertion" task where all performance is not very high.

---

### Official Review · Reviewer_GgNC · 2024-11-04

**Soundness:** 3
**Presentation:** 2
**Contribution:** 3
**Rating:** 6
**Confidence:** 3

**Summary:**

This paper introduces Stem-OB, a method that enhances visual imitation learning by using image inversion from pretrained diffusion models to reduce low-level visual variations while preserving high-level scene structure. Stem-OB creates a shared representation that is robust to various appearance changes without additional training. Empirical results on several benchmarks demonstrate the effectiveness in challenging environments with lighting and appearance changes.

**Strengths:**

* This paper focuses on enhancing the robustness of visual imitation learning, addressing a practical and impactful topic. The approach holds significant potential for advancing research in the field of robotics.

* The idea of using diffusion inversion to remove low-level visual variations while preserving high-level scene structures is both novel and intriguing.

* The evaluations are thorough, with experiments conducted in both simulated environments and on real-world robots. The real-world experiments highlight the method's strong generalization capabilities.

**Weaknesses:**

* I found the structure of this paper somewhat difficult to follow, as certain sections lacked clarity. For instance, in the preliminaries, the definition of diffusion inversion seemed to be a combination of both forward and backward diffusion. But in Figure 2, it seems that proposed method uses the noised observations as the policy input. If the proposed method is trained on the inversion-altered space, why using the noised version of the observation as input can improve the performance? How can the authors guarantee that applying forward diffusion to the image improves generalization?

* There seems to be a trade-off when choosing the inversion step, which can not be either too large or too small. Is there any explanation about this phenomenon?

* The theoretical analysis section was also challenging to understand. The authors discuss a loss between two latent variables,  $x_0$ and $y_0$. What  is the intuition of calculating the loss of two different images? From my understanding, the attribute loss here should refer to the inversed data $\hat{x}_0$ and its original version $x_0$. Could the authors also clarify the statement, “images with fine-grained attribute changes tend to become indistinguishable sooner than those with coarse-grained modifications under identical diffusion schedules”?

* Section 4.1 reminded me of another study [1], which employs diffusion models to purify noise within noisy demonstrations while preserving the optimal structure. Are there any conceptual similarities between that approach and the proposed method?

* Typos: Line 242: $er$f should be $erf$.

[1] Imitation Learning from Purified Demonstrations, ICML 2024.

**Questions:**

Please refer to Weakness.

---

> ### Author Response · Authors · 2024-11-21
> **Response to reviewer GgNC (1/2)**
>
> We thank reviewer GgNc for acknowledging our novelty and thorough evaluation! We'll address the concerns as follows:
>
> **Q1: Structure somewhat difficult to follow.**
>
> **A1:** Thank you for highlighting this point. We would like to clarify that diffusion inversion refers to a process where an original image is mapped to a noisy image while preserving semantic information to allow reconstruction during a separate backward process. The "inversion-altered space" consists of observations injected with inversion noise, which effectively emphasizes high-level structural information while erasing out lower-level details.
>
> During the inversion process, different images gradually lose their fine-grained details and converge towards Gaussian noise. the general idea of why an inversion-altered observation can improve the generalization performance is strongly related to our theoretical intuition: images that differ only in lower-level details become indistinguishable at an earlier inversion step. Consequently, observations of the same object under various visual perturbations converge to a unified "stem observation." Training policies on these unified observations enhance generalization to new environments, even before applying any downstream visual imitation learning (IL) pipeline.
>
> We have revised the relevant part in the preliminary part of the paper to better convey these concepts and improve clarity.
>
> **Q2: Trade-off between inversion steps**
>
> **A2:** This observation is indeed accurate, and the underlying reason aligns with our theoretical framework. As diffusion inversion merges different visual variations of the same scene, using too few inversion steps results in incomplete merging, which hinders the generalization ability. Conversely, excessive inversion may impact the semantic information critical to the task.
> In practice, we find a balanced inversion step of 15/50 to be effective for most real-world tasks and 10/50 for simulation tasks.
>
> **Q3: Theoretical analysis ambiguity**
>
> **A3:**
> As noted in the paper, the concept of attribute loss originates from [1]. Attribute loss quantifies the similarity between two images after a specified number of diffusion inversion steps, without imposing constraints on how these images are related. Accordingly, we adopt a general notation, $x_0$ and $y_0$, to represent two distinct images in Equation (6), consistent with the notation in [1]. A more formal definition of attribute loss is quantifying how likely DM fails to distinguish the noisy samples drawn from two overlapping noisy sample distributions $q(x_t|x_0)$ and $q(y_t|y_0)$ [1].
>
> In our context, the purpose of using attribute loss is to quantify the semantic differences between an image  $x_0$ and its variations $\hat{x}_0$ and $\tilde{x}_0$ with diffusion inversion step $t$. This is represented as $loss(x_0, \hat{x}_0, t)$ and $loss(x_0, \tilde{x}_0, t)$. We want to clarify that $\hat{x}_0$ and $\tilde{x}_0$ here are not the inversed data. They are source images without undergoing any inversion process. However, they exhibit some degree of similarity or relation to the original image $x_0$, which can be regarded as a deviation from $x_0$ in specific ways. For example, given a scene image $x_0$ shown in Fig. 4 (a), $\hat{x}_0$ indicates $x_0$ with only fine-grained attribute changes such as changing the texture on some of the objects. In contrast, $\tilde{x}_0$ indicates $x_0$  with coarse-grained modifications such as drastically changing the lighting condition of the scene. In this context, granularity refers to the pixel-level alterations introduced when modifying an attribute of the image. And what attribute loss indicates is $\hat{x}_0$ (fine-grained attribute change) becomes indistinguishable sooner than $\tilde{x}_0$ (coarse-grained attribute change) during diffusion inversion (equation (8)).
>
> [1] Yue, Z., Wang, J., Sun, Q., Ji, L., Chang, E. I., & Zhang, H. (2024). Exploring diffusion time-steps for unsupervised representation learning. arXiv preprint arXiv:2401.11430.

---

> > ### Author Response · Authors · 2024-11-21
> > **Response to reviewer GgNC (2/2)**
> >
> > **Q4: Relationship to another study**
> >
> > **A4:** We greatly appreciate you bringing this relevant study [1] to our attention. The mentioned study trains a diffusion process on optimal expert demonstrations and subsequently uses it to denoise partially re-noised sub-optimal demonstrations for purification. While their approach and ours share a key similarity —— both use a diffusion process to add noise and wipe out unwanted information while preserving high-level semantic, there are two main differences between the methods:
> >
> > 1. **Scope of Manipulation:** Their method applies the diffusion process to manipulate state-action transitions, aiming to edit trajectory information. In contrast, our approach focuses solely on modifying the observation space during the training phase.
> >
> > 2. **Reconstruction vs. Domain Merging:** Their approach includes a second denoising stage to purify noisy sub-optimal demonstrations, aligning them with the distribution of optimal ones. In contrast, Stem-OB is trained directly on the noisy observation space, aiming to merge observations from different domains rather than transferring between them.
> >
> > We agree that the comparison is informative, and we have added this study to the related works section in the revised manuscript.
> >
> > [1] Imitation Learning from Purified Demonstrations, ICML 2024.
> >
> > **Q5: Typo in line 242**
> > **A5:** Thank you for pointing this out! We've fixed it in the revised version. Due to a restructure of the paper, it's now located at line 249.

---

> ### Author Response · Authors · 2024-12-02
>
> Dear Reviewer GgNC,
>
> As the rebuttal period is coming to an end, we want to kindly check if there are any remaining questions or concerns regarding our responses to your review. We deeply value your feedback and are willing to provide any additional clarifications if needed.
>
> Thank you for your time and thoughtful assessment of our work!

---

### Author Response · Authors · 2024-11-21
**General Response**

We sincerely thank all the reviewers for their valuable and constructive comments. We greatly appreciate that most reviewers recognized our contributions. Specifically, all reviewers acknowledged that **the idea of using diffusion inversion to remove low-level visual variations while preserving high-level scene structures is novel yet straightforward**. Additionally, Reviewers GgNC, UAwF, and C5Ru agreed that **the empirical results, both in simulation and real-world settings, are thorough and convincing**. Building on this feedback, we address two common concerns raised by multiple reviewers: the theoretical underpinnings of our method and the variability in experimental results. While detailed explanations are provided in each reviewer's specific response.

### **Relation Between Theory and Method**

We would like to address the general confusion regarding the theoretical aspects of our work (Section 4.1), as raised by multiple reviewers.

The theory in our paper states that given two different images, these images will become indistinguishable earlier in the diffusion inversion process if their original representations are closer in the latent space of the diffusion model. It is important to clarify that "indistinguishable" in this context does not imply that the latent representations of the noisy images will be numerically close. Instead, it means that the distributions of the reconstructed images after inversion will overlap significantly, intuitively making it impossible for a human observer to infer which original image corresponds to the noisy image.

Building on this theoretical foundation, we conducted numerical analyses and user studies to validate that semantically similar images—such as objects within the same category but with variations in lighting, texture, or appearance—are indeed closer in the diffusion latent space. Moreover, human users were unable to distinguish between such images at an earlier stage of the inversion process. These findings justify our approach of using diffusion inversion to unify observations from similar tasks, thereby improving the generalization ability of downstream visual policies.


### **Experimental Results and Standard Deviation**

We would also like to address concerns regarding the large standard deviation observed in some of our experimental results. This variation arises because we report average success rates across all environment settings, encompassing both the training environment and several challenging test variants. While the policy demonstrates relatively high and consistent success rates in the training environment, its performance can vary significantly across the more difficult test variants. This aggregation naturally results in larger standard deviations.

To provide a clearer picture, we have separated success rates for the training environment and the test variants in our revised manuscript. For detailed statistics, including these breakdowns, please refer to our response to Reviewer UAwF and the appendix of the revised paper.

For all tables included in our responses and the revised manuscript, we **bold** the highest success rates and sometimes the second-highest when they are very close, for tasks of more than three baselines, we also ***bold-italicize*** the second-highest success rates.

### **Summary of Rebuttal Changes**
We conduct additional experiments to address the concerns of reviewers, a brief summary of the results includes:
1. (Reviewer UAwF) We add a new generalization setting: add six different normal maps to the tabletop along with its cross-over on two lighting conditions and two tabletop textures in the PushCube and PickCube tasks of the SAPINE environment. Stem-Ob still clearly outperforms all the baselines, showing its versatility over different generalization settings. (see Appendix H of the revised paper)
2. (Reviewer UAwF) We add four other tasks in the MimicGen environment with the same generalization settings as the existing four tasks. Stem-Ob performs better or comparable on these tasks than the best baseline method. (see Appendix F)
3. (Reviewer C5Ru) We add a self-supervised representation learning baseline R3M on PushCube and PickCube tasks of the SAPINE environment. Stem-Ob greatly outperforms it. (see Appendix I)
4. (Reviewer C5Ru) We test to perform inversion using another generative model: Stable Diffusion 1.4 on the MimicGen tasks, the results are comparable and sometimes better than current Stable Diffusion 2.1, showing our method's robustness. (see Appendix J)
5. (Reviewer UAwF) We conducted a cross-category generalization experiment demonstrating that Stem-Ob can generalize at higher levels of semantics. Performance improves as we increase the inversion step beyond what is used in intra-category settings, resonating well with our theory.

We also refactor the structure of the paper as suggested by reviewer pwGy, add a Problem Definition section, and make many other clarifications across the paper.

---

### Meta-Review · Area_Chair_ZjPH · 2024-12-21

**Metareview:**

This paper proposes Stem-OB, which suppresses low-level visual differences while maintaining high-level scene structures. The idea is to apply a diffusion inversion process to transform images into a shared image space without additional training. The approach is demonstrated with robust visual imitation learning. Given the compelling results and detailed description of the proposed approach, AC confirms that this paper can shed light on the possibility of the practical use of diffusion priors for visual imitation learning. However, due to image degradation, AC also sees that the fine details of the original image can be lost. The paper shows mainly the picking tasks for the large and few objects in the scene whose main structure is not affected much after the inversion. AC recommends extending the proposed idea for more complex manipulation cases that exhibit tiny and thin structures as future work.

**Additional Comments On Reviewer Discussion:**

In summary, all reviewers recommended the paper's acceptance, reaching a strong consensus. The common concerns raised by reviewers were the relation between theory and the proposed approach and questions regarding experimental results. The reviewers address the concerns carefully. The discussion between the reviewer UAwF and the authors was especially impressive, resulting in a significant improvement of the paper.

Specifically, the reviewer GgNC asks about the paper structure, the trade-off between inversion steps, theoretical analysis, and the relationship to another study. The author provided detailed feedback, but the reviewer did not respond and gave a rating of 6. The reviewer, UAwF, provided very detailed feedback and provided a strong score. The questions were about theory assumption, generalization, experiments, and standard variation. Authors provide feedbacks by providing extensive results on 50 episodes of SAPIEN and 300 MimicGen tasks. The reviewer UAwF and the authors have an extra discussion regarding the terms often used in the paper. The reviewer C5Ru asks to conduct a comparison with self-supervised representations and other generative models. The authors provided extra results, and reviewer C5Ru raised the score to 8. The reviewer pwGy gave a strong score with minor suggestions, and it is handled by the authors.

---

### Decision · Program_Chairs · 2025-01-22

Accept (Spotlight)